# TRACE: Contrastive learning for multi-trial time-series data in neuroscience

**Lisa Schmors**[1]    **Dominic Gonschorek**[2,3]    **Jan Niklas Böhm**[1]    **Yongrong Qiu**[7–9]
**Na Zhou**[4,5]    **Dmitry Kobak**[1]    **Andreas Tolias**[7–10]    **Fabian Sinz**[4–6]
**Jacob Reimer**[4,5]    **Katrin Franke**[2,7–9]    **Sebastian Damrich**[1]    **Philipp Berens**[1]

[1] Hertie Institute for AI in Brain Health, University of Tübingen, Tübingen, Germany
[2] Institute for Ophthalmic Research, University of Tübingen, Germany
[3] Centre for Integrative Neuroscience, University of Tübingen, Germany
[4] Department of Neuroscience, Baylor College of Medicine, Houston, USA
[5] Center for Neuroscience and Artificial Intelligence, Baylor College of Medicine, Houston, USA
[6] Institute for Computer Science and Campus Institute Data Science, University of Göttingen, Germany
[7] Department of Ophthalmology, Byers Eye Institute, Stanford University School of Medicine, USA
[8] Stanford Bio-X, Stanford University, USA
[9] Wu Tsai Neurosciences Institute, Stanford University, USA
[10] Department of Electrical Engineering, Stanford University, USA

## Abstract

Modern neural recording techniques such as two-photon imaging or Neuropixel probes allow to acquire vast time-series datasets with responses of hundreds or thousands of neurons. Contrastive learning is a powerful self-supervised framework for learning representations of complex datasets. Existing applications for neural time series rely on generic data augmentations and do not exploit the multi-trial data structure inherent in many neural datasets. Here we present TRACE, a new contrastive learning framework that averages across different subsets of trials to generate positive pairs. TRACE allows to directly learn a two-dimensional embedding, combining ideas from contrastive learning and neighbor embeddings. We show that TRACE outperforms other methods, resolving fine response differences in simulated data. Further, using *in vivo* recordings, we show that the representations learned by TRACE capture both biologically relevant continuous variation, cell-type-related cluster structure, and can assist data quality control.

## 1   Introduction

With advances in recording techniques, datasets in neuroscience have grown in size and complexity [7, 38]. For example, two-photon imaging and Neuropixel probes have made it possible to record responses of tens of thousands of neurons from multiple cortical areas under comparable conditions [12, 36]. To summarize and visually explore such noisy, high-dimensional data, it is invaluable to represent it in two dimensions to identify functionally similar groups of neurons [47].

A prominent paradigm for learning informative representations of data is contrastive learning [17, 29]. Here, representations are created by contrasting similar samples (referred to as "positive pairs") against dissimilar ones ("negative pairs"), ensuring that similar samples are grouped together, while dissimilar ones are separated. Contrastive learning has been popular for image data for some years [8], but has only recently seen first applications for neuroscience time-series data [35, 42]. For example, the contrastive method CEED [42] learns representations of extracellular action potential wave forms using generic data augmentations such as amplitude jitter. However, most contrastive learning methods, including CEED, do not directly embed into two dimensions necessary for visualization.

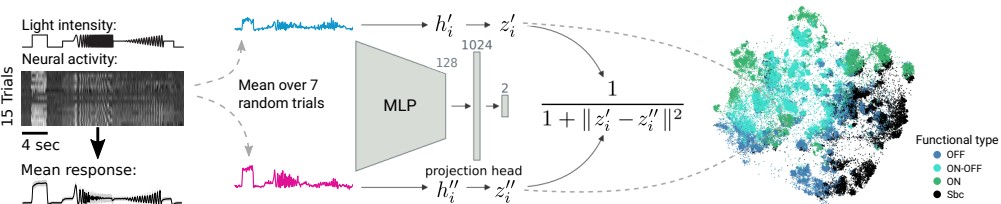

Figure 1: **TRACE embeds multi-trial time series using subset means for positive-pair generation.**
*Left:* Typical experimental structure in neuroscience: multiple trials of neural activity in response to
repeats of an identical stimulus (here, full-field light intensity modulations). For TRACE, positive
pairs are generated using means of subsets of trials. *Middle:* Positive pairs are fed through an MLP
and a fully-connected projection head to get representations $z_i'$ and $z_i''$. The loss function pushes $z_i'$
and $z_i''$ together and maximizes their similarity. *Right:* Final embedding of large-scale neuroscience
dataset recorded in superior colliculus. Color-coded according to their functional group.

Here, we exploit a common feature of neuroscience experiments – multiple recorded responses to
identical stimuli – to develop a new contrastive learning framework tailored for multi-trial time
series in neuroscience. This framework directly learns a two-dimensional (2D) representation of
the data. Building on $t$-SimCNE [6], we present TRACE: **T**ime series **R**epresentation **A**nalysis
through **C**ontrastive **E**mbeddings. Instead of using hand-tuned or generic data augmentations, TRACE
creates positive pairs by averaging across different subsets of trials, better capturing the structure of
time-series variation in neural responses (Fig. 1).

Using a synthetic dataset, we show that TRACE indeed automatically identifies informative parts of a
time series, where other methods are "drowning in noise". We then investigate the performance of
TRACE on a real-world two-photon-imaging and a Neuropixels time-series dataset. TRACE produces
visualizations that capture biological properties of the recorded cells better than other visualization
techniques. Our code is available at `https://github.com/berenslab/TRACE/tree/neurips25`
and `https://github.com/berenslab/TRACE_experiments`.

In summary, our contributions are:

1. We introduce a novel contrastive self-supervised framework for visualizing multi-trial time-series
data in neuroscience in 2D using the Cauchy similarity.
2. We develop a novel data augmentation technique for multi-trial time-series data based on averaging
subsets of trials to create positive pairs.
3. We establish the superior performance of our visualization method in terms of quality metrics and
alignment with biologically interesting cellular properties.
4. We show how the method can be used for identifying recording artifacts and investigating outliers.

## 2   Related work

Dimensionality reduction of neural data has been used in computational neuroscience for (1) assisting
the discovery of cell types or continuous functional variation and (2) finding low-dimensional
population dynamics. For (1), each point in the computed low-dimensional representation corresponds
to a neuron, while for (2), each point corresponds to a time point. Our approach addresses task (1).

Towards this goal, two of the most prominent non-linear visualization methods are $t$-SNE [41] and
UMAP [28]. While mostly used for single-cell transcriptomics data, they have also been employed
for wave-shape classification for electrophysiological data [23]. $t$-SNE and UMAP learn a non-
parametric 2D embedding, guided by pairs of nearest neighbors in data space. However, nearest
neighbors in the high-dimensional space can be a poor proxy for semantic similarity due to the curse
of dimensionality [18]. Alternatively, self-supervised learning approaches train encoder networks to
produce high-dimensional representations. For visual exploration, these need to be further reduced,
e.g. using PCA [32]. Instead of pairs of nearest neighbors, these methods rely on data augmentations,
better capturing the semantic similarities of samples. The type of augmentation depends on the data
modality, e.g. randomly resized crops and flips for images [8] or sampling consecutive time steps in

speech recordings [29]. Notably, $t$-SimCNE [6] uses data augmentations to create positive pairs of images but embeds into 2D to visualize the data.

Contrastive learning has been applied to medical time-series data, in particular electrocorticogram (ECoG) data. In addition to general time-series augmentations like scaling, blurring [34], and jittering [15], frequency- [49], cutout- [9], and permutation-based augmentations [34] have been used, which are not applicable to multi-trial non-periodic, stimulus-response data. Some of these works [13, 21] also create positive pairs based on patient identity. Mix-up [48] creates augmentations by blending different samples [4], which was also applied to ECoG data [9, 44]. Patient-based positive pairs or mix-up are similar in spirit to our approach. An important difference is that they only mix or pair two single trial samples and not a larger subset of trials, as we do. Our approach balances positive pair variability and similarity to inference-time inputs.

In neuroscience, there has been comparatively little work on contrastive learning for time-series representations with the goal of identifying discrete cell types (i.e. clusters). Peterson et al. [33] use a self-supervised model to generate pseudo labels and enrich the collected data this way. Cho et al. [10] suggest a time-warping loss, which is not applicable to our experiments, which are aligned in time by construction. Related to our approach, the method CEED has been used for spike sorting and cell-type classification [42]. CEED generates representations using generic time-series as well as task-specific augmentations such as spike collision and channel subset selection. As is common for contrastive self-supervised learning, CEED does not embed into two dimensions directly, but relies on an additional dimensionality reduction step for a 2D visualization, unlike our method. Moreover, TRACE captures the local noise structure better than CEED's generic augmentations (Fig. 2).

For learning representations of neural population dynamics, methods like LFADS [31], Swap-VAE [26], or the contrastive approaches MYOW [3], CEBRA [35], SinkDivLM [40], and Neuroformer [2] produce embeddings in which the temporal evolution of the population activity and behavioral modulation of neural activity can be explored. Here, each embedding point corresponds to a time point, not a single-neuron response like in TRACE. Thus, they tackle a fundamentally different problem and are not applicable in our setting (see point (2) in the beginning of this section).

Similarly, many general-purpose contrastive time-series models, like TNC [39], also produce time-point embeddings and are thus not applicable in our setting. Others, such as TS2Vec [46] contrast instances of time series, but learn high-dimensional embeddings, not geared towards visualization. We find that TRACE outperforms TS2Vec even when adapting the latter to a visualization setting.

NEMO [45] and PhysMAP [24] contrast different electorphysiological modalities, while TRACE works by contrasting multiple recordings of the same modality.

## 3 TRACE: contrastive learning for multi-trial time-series data in neuroscience

Our goal is to create informative 2D embeddings of multi-trial time-series data from neuroscience experiments, supporting data exploration, cell-type discovery, and other clustering tasks. To do so, we developed TRACE, a method that makes use of the trial-based nature of many neuroscience experiments to create positive pairs by averaging over subsets of trials. It builds on the self-supervised method $t$-SimCNE [6], which is based on the SimCLR framework [8] and directly learns a 2D embedding. We briefly review contrastive learning and $t$-SimCNE before describing our approach. Finally, we will briefly describe the contrastive learning frameworks CEED and TS2Vec as our closest competitors using generic data augmentations for time series [42].

### 3.1 Contrastive self-supervised visualizations

Given a dataset in a data space $X$, contrastive self-supervised learning trains the parameters $\theta$ of a neural network $f_\theta : X \to Z$ to obtain representations $z = f_\theta(x)$ in embedding space $Z$. It learns salient features by making the embedding invariant to known data similarities encoded in positive pairs, which are typically obtained as two modifications $x'$, $x''$ of the same data sample $x$. Similarity in the embedding space is measured by a similarity function $q : Z \times Z \to \mathbb{R}$, Eqs. (2, 3). Positive pairs $(x', x'')$ should have high similarity $q(z', z'')$, i.e. $z'$ and $z''$ should be close. This is typically

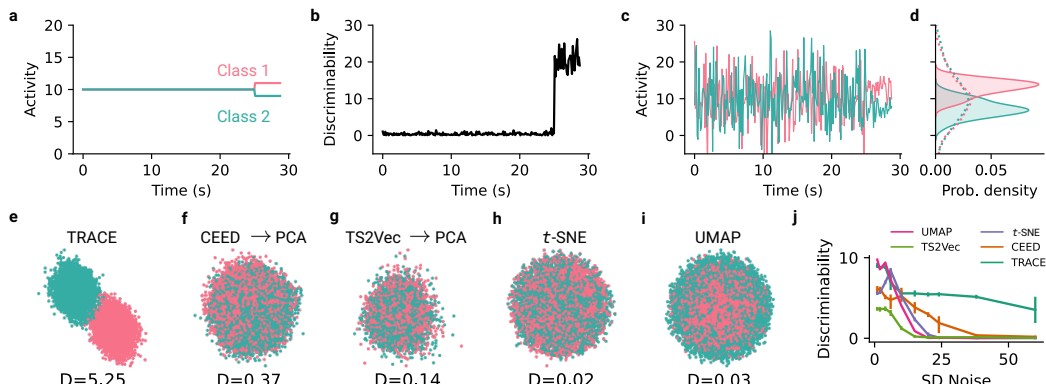

Figure 2: **Synthetic dataset shows ability of TRACE to identify cell types in noisy data. (a)** Simulation of two classes of neural responses with baseline (25 s) and type-specific response (5 s) periods. In the baseline period, both classes show 10 Hz activity, while in the type-specific response period they fire at 11 Hz and 9 Hz, respectively. Responses are corrupted by variable amounts of noise (Sec. 4.1). **(b)** Discriminability [14] of the two neuron classes per bin was measured using $D = \frac{|\mu_1 - \mu_2|}{0.5(\sigma_1 + \sigma_2)}$ (mean across neurons shown). **(c)** Mean neural responses across 10 trials for two example neurons from the two types. **(d)** Distributions of baseline (dashed) and response periods (solid) for the two neuronal responses in (c). **(e–i)** Embeddings of 10k simulated neurons for baseline activity noise SD = 38 with 10 trials each. **(j)** Discriminability for simulated responses with increasing amounts of noise during the neural baseline activity. Error bars indicate 95% confidence intervals.

achieved with the InfoNCE loss, which for the $i$-th positive pair $(x_i', x_i'')$ of the training batch is

$$\mathcal{L}(x_i', x_i'') = -\log \frac{q(z_i', z_i'')}{q(z_i', z_i'') + \sum_{\alpha \neq i} \left( q(z_i', z_\alpha') + q(z_i', z_\alpha'') \right)}. \tag{1}$$

Here, $\alpha$ runs over all other positive pairs in the training batch. The pairs $(z_i', z_\alpha')$ and $(z_i', z_\alpha'')$ are called negative pairs and their similarity is decreased, i.e. $z_i'$ and $z_\alpha'$ (or $z_\alpha''$) pushed apart.

Usually, the embedding space is constrained to a high-dimensional hypersphere $Z = \mathbb{S}^{d-1} \subset \mathbb{R}^d$ and the similarity function for $z, \tilde{z} \in Z$ is based on the cosine similarity:

$$q(z, \tilde{z}) = \exp \left( (z^\top \tilde{z}) / (\|z\| \|\tilde{z}\| \tau) \right), \tag{2}$$

where the temperature $\tau$ is a hyperparameter. However, we need 2D representations for visualization. Setting $d = 2$ would result in embedding everything into a circle, which is unsuitable for this purpose.

$t$-SimCNE [6] learns a 2D visualization of image datasets with a parametric encoder and data augmentations as similarity source. The key ingredient is the $t$-SNE-inspired Cauchy kernel

$$q(z, \tilde{z}) = (1 + \|z - \tilde{z}\|^2)^{-1} \tag{3}$$

in the 2D embedding space $Z = \mathbb{R}^2$. We adapt $t$-SimCNE to multi-trial time-series data from neuroscience by implementing positive pairs based on trials, replacing the original image-based transformations (see below). We adopt a simpler (one-stage) training procedure compared to $t$-SimCNE, directly learning the 2D embedding.

## 3.2 TRACE uses subset means as positive pairs

In many neuroscience experiments, the activity of a set of neurons is recorded repeatedly under identical stimulation conditions, e.g. presenting a visual stimulus multiple times (Fig. 1, left). We denote the time series of the $l$-th trial for neuron $i$ as $x_i^l[t]$, with $t \in \{1, \dots, T\}$ indexing discrete time. For all data points we had the same number of time steps, so we omit the time index $t$ for clarity.

We reasoned that the variation between responses in different trials provides an estimate for the naturally occurring variability at each time point, e.g. due to fluctuating brain state or inaccuracies in the measurement. This is precisely the type of noise to which the embedding should be invariant.

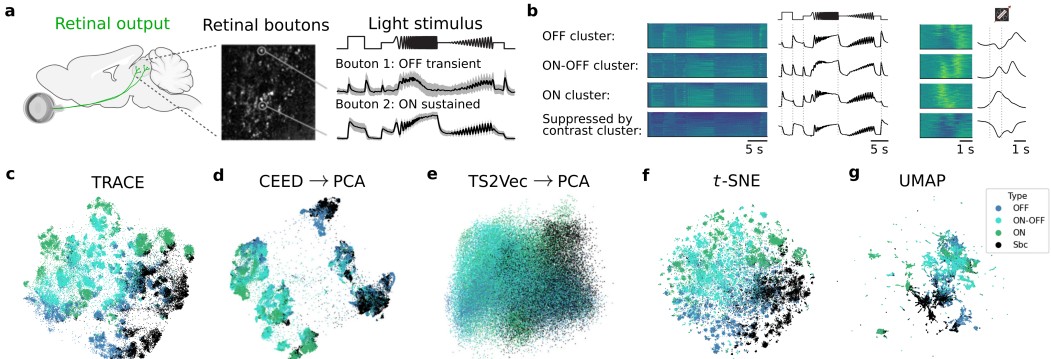

Figure 3: **Neural dataset from retinal ganglion cells measured in superior colliculus and its embedding across different methods**. **(a)** Schematic of recording neural activity. Each neuron projecting to superior colliculus (*left*) has many retinal boutons (*middle*) from which we record activity in response to a full-field light stimulus using two-photon calcium imaging (*right*). **(b)** We functionally clustered the light-evoked neural activity into four major groups (OFF, ON-OFF, ON, Suppressed-by-contrast, Sbc) showing their mean response to the "chirp" (*left*) and the "moving bar" stimulus (*right*). **(c)** TRACE embedding of the neuronal time series colored by their functional group, **(d, e)** Same as (c), but for CEED / TS2Vec visualized in 2D using PCA, **(f)** *t*-SNE, and **(g)** UMAP.

Therefore, we use averages of two random, non-overlapping subsets of the trials as positive pairs, making the representation invariant to the trial-to-trial fluctuations. Formally, let $r$ be the total number of trials, which are randomly split into two equisized, non-overlapping subsets $S_1$ and $S_2$ with

$$S_1 \cap S_2 = \emptyset \quad \text{and} \quad |S_1| = |S_2| = k \leq r/2. \tag{4}$$

Then the positive pair of neuron $i$ consists of the means of these non-overlapping trial subsets:

$$x_i' = \frac{1}{k} \sum_{l \in S_1} x_i^l \quad \text{and} \quad x_i'' = \frac{1}{k} \sum_{l \in S_2} x_i^l. \tag{5}$$

The subsets $S_1$, $S_2$ are dynamically resampled per neuron in each epoch. A training batch is made up of the positive pairs of that batch's neurons. This approach is inspired by mix-up augmentations [48], but generalized to averages of more than two samples and restricted to mixing only trial responses of the same neuron. The number $k$ of trials to average over is a hyperparameter, but in practice we use $k = \lfloor r/2 \rfloor$ as it provides a good trade-off between variability of the positive pairs, and similarity to the full mean which we embed at inference time. This choice also leads to the largest number of distinct positive pairs, $\binom{r}{\lfloor r/2 \rfloor}$. The subset means from other neurons in the batch were used as negative pairs.

### 3.3 Alternative contrastive learning framework for embedding time series

We compare against two other contrastive time-series models, CEED [42] and TS2Vec [46]. CEED provides the closest alternative to our framework, as it uses contrastive learning with general purpose time-series data augmentations and task-specific data augmentations, applicable to extracellular action potential wave forms [42]. To compare with CEED in the context of general neuroscience time series, we re-implemented their general purpose time-series augmentations: (1) amplitude jittering, (2) temporal jittering, and (3) correlated background noise. For more details see Appendix A.7. CEED embeds time series into five-dimensional (5D) space, which is mapped to 2D using PCA for visualization (alternative UMAP results are provided in Fig. S6, Table S3).

TS2Vec employs two types of contrastive losses: In the first, it contrasts representations of different time series (like CEED and TRACE). In the second, it contrasts representations of different time points from the same time series. These losses are applied through a hierarchical process — starting with two time segments and progressively subdividing it into smaller units — enabling both inter-series and temporal intra-series contrasting at multiple scales. TS2Vec outputs in $\mathbb{R}^{320}$ and uses the cosine similarity in its loss function, like CEED. When applying CEED and TS2Vec to our neural data, we

use the mean neural response across trials. Both methods apply identical transformations for each recorded neuron $i$. In contrast, our approach of using the means across a subset of trials allows to automatically identify the local noise structure for each neuron (Sec. 3.2).

We compared our 2D TRACE embeddings with the default versions of CEED and TS2Vec using PCA (as suggested by the CEED paper) to obtain a 2D visualization of their 5D and 320D outputs, respectively. We call these CEED → PCA and TS2Vec → PCA (Fig. 3d,e). We also ablated TRACE's two main contributions: generating positive pairs by averaging across a subset of trials and Cauchy similarity on 2D outputs. By modifying CEED and TS2Vec to output 2D embeddings with Cauchy similarity, we created hybrid models, TRACE + CEED and TRACE + TS2Vec (Fig. S3, App. A.6).

## 4 Experimental setup

### 4.1 Datasets

**Synthetic dataset**   We simulated artificial neuronal responses of two cell types with distinct temporal structure: Each response consisted of 25 seconds of baseline activity followed by 5 seconds of class-specific signal, with either positive or negative amplitude defining the two cell types (Fig. 2a–d). To mimic realistic recording conditions, we added Gaussian noise with higher variance during the baseline activity (amplitude 10, standard deviation (SD) from 1 to 60 for single trial) and lower variance during the class-specific response period (amplitude 9 or 11, SD 8 for single trial), resembling known effects of stimulus onset in visual neurons [11]. We generated a typical number of 10 trials per neuron. While idealized, two neuron types may indeed differ in their response to only part of the stimulus, making long response periods uninformative or even detrimental for separating them.

**Large-scale neural calcium imaging dataset**   We used a large-scale two-photon-imaging dataset from *in vivo* mouse retinal ganglion cell axon endings, measured in superior colliculus of awake, head-fixed mice (Fig. 3a). These neurons expressed the genetic calcium indicator GCaMP8m under the hSyn promoter. We presented two visual stimuli to the animal during recordings: (1) a full-field "chirp" consisting of a bright step and two sinusoidal intensity modulations (Fig. 1), and (2) local bright moving bars on a dark background in eight directions (Fig. 4f, right). In total, the dataset consisted of recordings of $71,021$ individual retinal ganglion cell boutons measured at a sampling frequency of 8 Hz, leading to time series with 260 time bins. For the two stimuli, 15 and 10 repeated trials were recorded, respectively. We manually split responses into four groups using the responses to the moving bar stimulus (see App. A.1, Fig. 3b): ON, OFF, ON-OFF, and Suppressed-by-contrast (Sbc) and to evaluate the cluster structure of the embeddings we clustered the data using a Gaussian mixture model with 50 components. All procedures were approved by the the Baylor College of Medicine, Houston, USA, animal protocol number: AN-8132.

**Allen Institute Neuropixels spiking dataset**   We used the Allen Institute Neuropixels "visual coding" dataset [36], which is part of the Allen Brain Observatory to test performance on a Neuropixels spiking dataset with action potential resolution. The dataset recorded activity of single neurons across visual cortical and thalamic structures in awake, head-fixed mice viewing diverse visual stimuli using Neuropixels silicon probes. After quality filtering and excluding non-visual neurons, we analyzed spiking responses of $10,322$ neurons to light flashes and drifting gratings across visual brain areas (for details see App. A.5). To compute the *ARI score* and *kNN accuracy*, we used the labels of brain area. Biologically meaningful metrics provided by the Allen Institute were: orientation selective index (OSI), preferred orientation (PO), grating modulation ratio (F1/F0), natural image selectivity (NIS; based on responses to natural images not used to create the embeddings), behavioral modulation (correlation of firing rate with running speed of the mouse; not used to create the embeddings). Alignment of the embedding with biological metrics was measured with *kNN regression* ($R^2$). For PO and behavior, we computed the radial correlation, as it better captured their global structure in the embedding.

### 4.2 Quantitative measures of embedding quality

We evaluated the embedding quality using different metrics:

1. The *ARI score* [20] between response groups identified in the 2D embedding using Mixture of Gaussian clustering and either the group labels of the calcium imaging dataset or brain region of

Table 1: **Quantitative model performance the large-scale superior colliculus dataset**. The columns are: model type (see text), the ARI score, $k$NN accuracy, Spearman correlation ($r_S$), the maximum correlation for ON-OFF-index ($r_{OOi}$), response transience index ($r_{RTi}$), the recording depth $r_{Depth}$, and the average rank $\mu_{Rank}$ of each method. For a definition of the measures, see Sec. 4.2. All metrics other than the rank are better when higher. Uncertainties for the ARI score were insignificant and we omitted them. The best values for each metric are **bold**.

| Model | ARI | $k$NN acc. | $r_S$ | $r_{OOi}$ | $r_{RTi}$ | $r_{Depth}$ | $\mu_{Rank}$ |
|---|---|---|---|---|---|---|---|
| TRACE | **0.28** | $69.7 \pm 0.3\%$ | $0.45 \pm 0.01$ | $0.67 \pm 0.02$ | $\mathbf{0.26 \pm 0.00}$ | $0.54 \pm 0.02$ | **2.17** |
| + CEED | 0.25 | $64.0 \pm 0.4\%$ | $0.51 \pm 0.02$ | $0.64 \pm 0.01$ | $\mathbf{0.26 \pm 0.00}$ | $0.48 \pm 0.04$ | 2.67 |
| + TS2Vec | 0.10 | $23.8 \pm 1.1\%$ | $\mathbf{0.60 \pm 0.01}$ | $0.28 \pm 0.04$ | $0.23 \pm 0.06$ | $0.51 \pm 0.00$ | 4.33 |
| CEED $\rightarrow$ PCA | 0.20 | $60.7 \pm 0.6\%$ | $0.45 \pm 0.05$ | $0.54 \pm 0.10$ | $0.17 \pm 0.00$ | $\mathbf{0.55 \pm 0.07}$ | 4.00 |
| TS2Vec $\rightarrow$ PCA | 0.04 | $11.6 \pm 0.9\%$ | $0.30 \pm 0.01$ | $0.12 \pm 0.01$ | $0.09 \pm 0.01$ | $0.24 \pm 0.01$ | 6.83 |
| $t$-SNE | 0.24 | $\mathbf{70.7 \pm 0.2\%}$ | $0.50 \pm 0.01$ | $\mathbf{0.69 \pm 0.01}$ | $0.17 \pm 0.02$ | $0.42 \pm 0.03$ | 3.00 |
| UMAP | 0.23 | $69.9 \pm 0.2\%$ | $0.27 \pm 0.02$ | $0.38 \pm 0.05$ | $\mathbf{0.26 \pm 0.02}$ | $0.42 \pm 0.00$ | 4.00 |

the Neuropixels dataset, respectively. This quantifies how well the 2D embedding captured major response groups.

2. The *kNN accuracy* as a standard metric that quantifies embedding quality by predicting the response-type labels through a majority vote of each point's 15 nearest neighbors in the 2D embedding space. This metric indicates how well points from the same response group cluster together and is commonly used in contrastive learning [30].

3. *Spearman's rank correlation* $r_S$ [37] between pairwise distances in the original time-series space and the low-dimensional embedding, $r_S = \text{corr}(\|x_i - x_j\|, \|z_i - z_j\|)$ to quantify how well the embedding preserved the relative distances between data points. This is a common metric for visualization methods [22], but rests on the assumption that distances between the original data are meaningful, which may not be the case due to the curse of dimensionality ([1]).

4. The *linear or radial correlation* $r$ between the 2D embedding and biological relevant variables such as the ON-OFF index ($OOi \in [-1, 1]$), the response-transience index ($RTi \in [0, 1]$) (see App. A.2, A.3), and the recording depth for the calcium imaging dataset (Fig. S4) and preferred orientation (PO) and behavioral modulation for the Neuropixels dataset. For linear gradients, we first fit a linear regression between the embedding coordinates $(x, y)$ and each variable to determine the rotation angle $\theta$ that maximized correlation along one axis. The radial distance of each point was determined based on its Euclidean distance from the arithmetic mean of the embedding. Absolute Pearson correlations were computed between biological variables and both the $\theta$-direction (linear) and radial distances and the maximum reported in Table 1.

5. The *kNN regression* to measure how well the respective biological variable of a neuron is predicted by averaging the values of its $k = 15$ nearest neighbors in the embedding (used for the Neuropixels dataset as it sometimes better captured the global structure in the embedding).

6. The *average rank* of the above metrics to provide an aggregated score.

## 5   Results

The goal of TRACE is to support exploratory data analysis of multi-trial time-series data in neuroscience to identify functional cell types or groups and to study continuous variation of neuronal function between cells. Here, we show that TRACE performs better than its competitors both on a synthetic dataset (Fig. 2) and on two large-scale *in vivo* neuroscience time-series datasets (Figs. 3,S8).

### 5.1   TRACE identifies informative regions in a synthetic dataset

In the synthetic dataset, responses of two artificial neuron types were constructed such that the first 25 seconds corresponded to baseline activity with high noise, while the final 5 seconds distinguished two neuron types with high signal-to-noise-ratio (Fig. 2a–d, Sec. 4.1). We compared TRACE against CEED, TS2Vec, $t$-SNE, and UMAP and found that TRACE successfully separated the two neuronal types, revealing differences in responses that the other methods failed to distinguish (Fig. 2e–i).

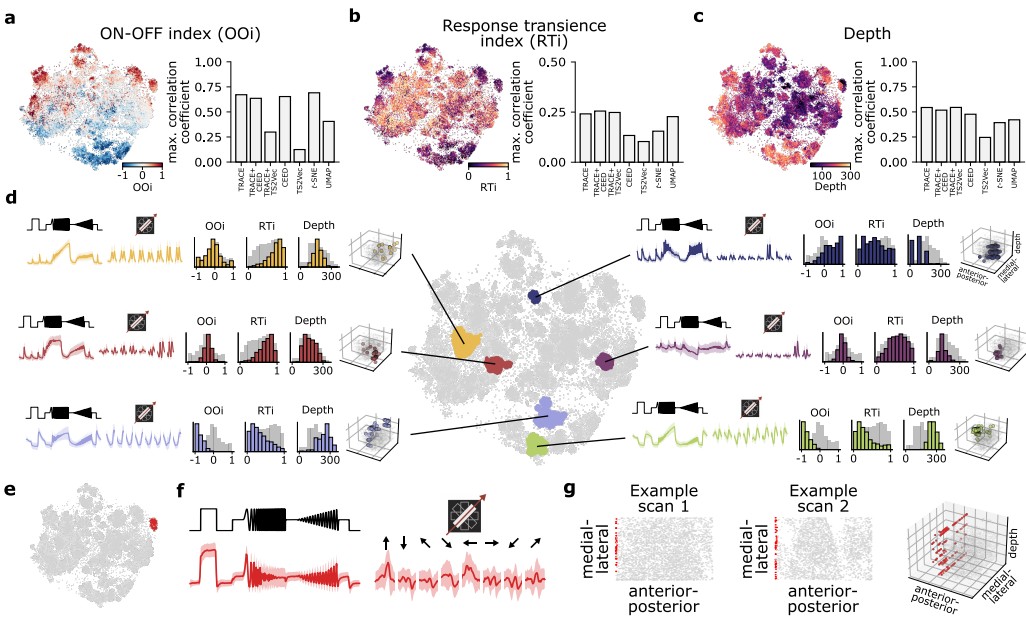

Figure 4: **The TRACE embedding reflects variables of biological interest and aids in identifying experimental artifacts.** **(a)** *Left*: TRACE embedding color-coded by the ON-OFF index (OOi). *Right*: Maximum correlation with OOi across all methods. **(b)** Same as (a), but for the response transience index (RTi), and **(c)** the recording depth in the superior colliculus. **(d)** TRACE embedding with example islands extracted using HDBSCAN clustering ($\epsilon = 0$, minimum numbers of samples per cluster 50, minimum cluster size 500). Each example shows (from *left* to *right*) the mean response to the chirp and the moving bar stimulus, the OOi, the RTi, the recorded depth and the anatomical location for the respective cluster. **(e)** TRACE embedding with artifact island (red). **(f)** Responses of the artifact island to the chirp (*left*) and the moving bars (*right*). **(g)** Two example scan fields with neuronal responses identified as artifacts in red and others in gray, as well as anatomical location across the entire recording volume.

TRACE even performed well when using a smaller number of trials $k$ used for the non-overlapping subsets (Fig. S2a). In addition, we varied the noise levels by increasing the baseline standard deviation. TRACE outperformed all other methods and was able to successfully separate the two classes even at high noise levels (Fig. 2j, S1, Table S1). The other methods performed well at low noise but their performance rapidly deteriorated as noise increased.

To successfully separate the two classes, the methods needed to learn to ignore the first period of spontaneous activity. $t$-SNE and UMAP struggled because they compute distances between responses and these are dominated by uninformative distances induced by the spontaneous activity (Fig. 2c). In CEED, only the correlated background noise can have varying effects across different time bins. However, as the variance of the noise is estimated on the whole dataset without knowledge of the true label, the variance is inflated in the signal part of the time series because of inter-class variability. As a result, CEED's covariance augmentation might transform a sample with positive response amplitude into one with negative response amplitude, destroying the class structure. In contrast, the sample specific subset mean augmentation used by TRACE does not have this problem: The subset means for a sample with positive response amplitude will retain this positive signal response, and only differ strongly on the low signal-to-noise part of the recording. Thus, TRACE specifically learned to ignore this part of the time series and only focuses on the part that encodes the different classes, explaining why it can produce a visualization with clearly separated classes (Fig. 2e, S1a).

## 5.2 TRACE represents biological structure better in superior colliculus two-photon dataset

Next, we applied TRACE to two-photon calcium recordings of retinal ganglion cell boutons measured in the mouse superior colliculus responding to light stimuli (Fig. 3a, Sec. 4.1 for details). Based on the recorded time series, we identified four major neural response groups based on the responses to the

moving bar stimulus: ON, OFF, ON-OFF, and Suppressed-by-contrast (Sbc) neurons (Fig. 3b). We compared TRACE against CEED, TS2Vec, $t$-SNE, and UMAP (Fig. 3c–g, Table 1) and additionally extended the TRACE setup to use either CEED-like augmentations or the TS2Vec approach (see Sec. 3.3, Fig. S3). We found that the embedding learned by TRACE and TRACE-variants visually exhibited the best balance between resolving cluster structure and retaining large-scale structure (Fig. S3). This was confirmed quantitatively: For example, TRACE most accurately reflected the manually identified response groups (ON, OFF, ON-OFF, and Suppressed-by-contrast), as indicated by the best ARI score (Table 1). In addition, TRACE showed comparable $k$NN accuracy to $t$-SNE, in contrast to CEED or TS2Vec, indicating that nearest neighbour in the embedding typically came from the same neuronal response group. Finally, TRACE and TRACE-variants showed the best correlation between time-series distances and embedding distances, showing that the embedding overall respected the structure of the high-dimensional space well. Computationally, TRACE was significantly faster than TS2Vec (Table S5) and more efficient than CEED. While CEED needs expensive data augmentations for each observation in the batch we pre-computed 10k noise samples to improve efficiency (not counted in the reported training time). This is not necessary for TRACE as the mean of a subset of trials is used for the positive pairs. Testing how reducing the number of trials ($k$) in non-overlapping subsets affects the results we found that TRACE still achieved good performance even with $k = 1$ (Table S2, Fig. S2b).

We next studied to what extent the different embeddings captured other neuronal properties such as the tendency to respond to light increments or decrements (measured by the ON-OFF index (OOi), Fig. 4a) or the kinetics of the response to a light step (measured by the response transience index (RTi), Fig. 4b) [5]. We found that TRACE and TRACE-variants captured these properties well or better than competing methods, yielding the highest correlation with RTi and comparable correlation values for OOi and depth (Table 1, Fig. 4a–c, Fig. S4). Notably, the depth is a completely independent measure never used during learning (whereas OOi and RTi were derived from the shape of the time-series activity used for training). When ranking methods by their mean rank ($\mu_{\text{Rank}}$) across all evaluation metrics, TRACE came first among the evaluated models showing that it consistently performs well across all metrics, in line with the visual impression (Table 1, Fig. 3, Fig. S3). TRACE also performed competitively in higher embedding dimension (Table S4).

Next, we clustered the 2D TRACE representation using HDBSCAN [27] to explore the structure of the visualization in more detail (Fig. 4d). We found that distinct clusters in the embedding showed unique neural response characteristics, representing types within the manually defined response groups of ON, OFF, ON-OFF, and Sbc (examples shown in Fig. 4d). Interestingly, many of these identified subgroups clustered in specific, spatially distinct regions within the superior colliculus, suggesting that the TRACE embedding learned the known relationship between neural responses and anatomical location (without having access to the location during training) [25].

## 5.3 TRACE finds recording artifacts and outliers

In the TRACE embedding, we found an isolated island on the far right, which showed peculiar light responses (Fig. 4e), closely following the light intensity of the stimulus (Fig. 4f). We found that these responses were exclusively recorded on the far left of a scan field (Fig. 4g), suggesting that they corresponded to light artifacts. In the representations for TRACE + CEED and CEED this island was also visible but less clearly separated. Interestingly, the only other embedding clearly showing these artifacts as outliers was that of UMAP, while they did not stand out at all for TS2Vec (Fig. S5).

Finally, there were a few outlier points located in the white space between clusters, which did not seem to belong to any of the clusters (Fig. S7). To detect outliers we identified outliers as points that are both locally isolated (few neighbors) and globally sparse (low density) (App. A.4). We investigated these neural responses and found some obvious examples of outlier responses that show noisy responses for either of the two stimuli, suggesting TRACE can be used for data cleaning.

## 5.4 TRACE shows superior performance on Neuropixels spiking dataset

Next, we tested the generalizability of our approach across domains and applied it to the Allen Institute Neuropixels dataset [12, 36] that included diverse responses of visual neurons from different brain areas to visual stimuli such as dark and bright flashes and drifting gratings (Fig. S8a).

We evaluated embeddings using distance correlation $r_S$ and clustering metrics (ARI score, $k$NN accuracy) with brain region as ground truth labels. While clear regional separation is not necessarily expected because hierarchical visual processing naturally creates overlapping response patterns across areas, the brain region provided the only available categorical ground truth for this dataset (Fig. S8b,c). We therefore additionally evaluated embeddings using biologically meaningful continuous variables (OSI, PO, F1/F0; see methods and Table S6). To test generalization beyond the stimulus features used to create embeddings, we also used the natural image selectivity (NIS) and behavioral modulation (correlation with running speed) to evaluate the embedding structure, neither of which were derived from the flash and grating responses used for creating the embeddings (Fig. S9).

While standard methods ($t$-SNE, UMAP) found some structure especially in terms of distance correlation and for OSI, TRACE outperformed them in most metrics and outperformed CEED in all metrics (Table S6; Fig. S8d–h). The performance of TS2Vec variants was mixed, with good preservation of some structure (OSI, NIS) but very poor global distance correlation (Fig. S9). Overall, TRACE achieved the best rank by far, indicating best overall performance on this dataset (Table S6). Notably, while TRACE was the only methods capturing both image selectivity and modulation by behavior well (despite these features not being used for training), only TRACE was able to capture this and produce clearly structured, interpretable embeddings (Fig. S9).

# 6 Conclusion, limitations, and future work

We presented TRACE, a new framework that combines contrastive learning with neighbor embeddings to directly generate interpretable 2D visualizations of large-scale neural time-series data. Using the inherent structure of multi-trial recordings common in neuroscience experiments, TRACE is able to separate subtle differences of simulated neural response types that competing methods fail to distinguish. When applied to a diverse neural dataset of two-photon recordings, TRACE captured both continuous variations in neural properties and discrete cell-type structures, and identified clusters with fine differences in functional responses and highlighted recording artifacts in the learned 2D representation (Sec. 5.3). TRACE proved especially valuable for Neuropixels spike train data, because its approach of trial averaging preserves the underlying Poisson statistics better than standard augmentations used by other contrastive methods (e.g. temporal jitter or additive Gaussian noise).

Conceptually, TRACE has two advantages over existing methods. First, methods like $t$-SNE and UMAP based on neighborhood relationships in the high-dimensional space are sensitive to uninformative, noisy baseline activity and may not be able to detect short class-informative response periods (Sec. 5.1). Contrastive frameworks such as CEED rely on general-purpose data transformations and may mask subtle response differences between cell types. Second, TRACE directly produces a 2D visualization while other contrastive methods require a separate reduction step. As a result, the embeddings produced by TRACE were the most biologically interpretable as revealed by their ability to reflect key response characteristics (Sec. 5.2).

A limitation of our work is the need to train an embedding model, increasing computational time compared to $t$-SNE or UMAP. However, this comes with the benefit of improved embedding quality and the ability to map new data points. Another limitation of our work is the need for repeated trials. However, this data collection approach data is standard for many neuronal time-series datasets. In some cases, between-trial variation may be of interest, e.g., when investigating adaptation over multiple trials. TRACE operates under the assumption that differences between trials are undesirable noise thus making it not the right tool for this type of question.

In future work, it will be interesting to apply TRACE to a much wider range of time-series data because many non-neuroscience datasets also have inherent multi-trial structure or can be reshaped into this format. For example, in sport analytics inertial measurements are often taken during repeated exercises, such as basketball free-throw drills [16, 19]. The the Google Speech Command Dataset [43] contains 5 audio recordings per speech commands and speaker. For medical data such as electrocardiography one could use different daily cycles or stress tests as trials. For financial market data, trading periods can be used as trials to detect unusual patterns or market anomalies. Another application domain could be climate data where years, seasons, or tidal cycles could be used as trials. These avenues highlight the broad applicability of leveraging multi-trial structure with TRACE.

## Acknowledgments

The authors thank the Hertie Foundation, the German Research Foundation (CRC 1233 "Robust Vision" and DFG/ANR EU 42/12-1, Project number 505379160) and the International Max Planck Research School for Intelligent Systems (IMPRS-IS) for financal support. FS acknowledges the support of the Lower Saxony Ministry of Science and Culture (MWK) with funds from the Volkswagen Foundation's zukunft.niedersachsen program (project name: CAIMed-Lower Saxony Center for Artificial Intelligence and Causal Methods in Medicine; grant number: ZN4257. KF was supported by the European Research Council (Starting Grant "Eye To Action" 101117156). PB and DK are members of the EXC 2064 "Machine Learning — New Perspectives for Science".

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

# A  Experimental details

## A.1  Identifying neural response groups for the calcium imaging dataset

A typical coarse classification of retinal ganglion cells is by their response to On- and Off-sets in the light stimulus and can typically be obtained with high confidence from simple statistics of the mean trial response of the moving bar stimulus [5]. In particular, suppressed-by-contrast responses were identified as the ones with negative area under the curve, calculated from their mean response across all directions of the moving bar. ON-OFF responses were identified using the characteristic double-peak (2nd component in PCA captured this double-peak). Finally, ON and OFF responses were classified based on their response direction (captured in 1st component of PCA).

## A.2  ON-OFF index

The ON-OFF index (OOi $\in [-1, 1]$) was calculated to quantify the response polarity of the recorded visual cells to the "chirp" light stimulus. Increased activity during light increments would indicate a response characteristic as 'ON' polarity (OOi $\sim 1$), increased activity during light decrements as 'OFF' polarity (OOi $\sim -1$), and a response to increments and decrements as 'ON-OFF' polarity (OOi $\sim 0$).

For a mean response $x$, the ON-OFF index (OOi) was computed as

$$\text{OOi} = \frac{r_{\text{ON}} - r_{\text{OFF}}}{r_{\text{ON}} + r_{\text{OFF}}}, \tag{6}$$

where $r_{\text{OFF}}$ and $r_{\text{ON}}$ are defined as the integrated response over the short time interval ($\Delta = 1\,\text{s}$) after the start ($t_{\text{ON}} = 2.5\,\text{s}$) and the end ($t_{\text{OFF}} = 5.5\,\text{s}$) of the initial light stimulus during the "chirp":

$$r_{\text{ON}} = \sum_{t=t_{\text{ON}}}^{t_{\text{ON}}+\Delta} x[t] \quad \text{and} \quad r_{\text{OFF}} = \sum_{t=t_{\text{OFF}}}^{t_{\text{OFF}}+\Delta} x[t] \tag{7}$$

Then, metric was clipped within the range $[-1, 1]$ to obtain a bounded metric.

## A.3  Response transience index

The response transience index (RTi $\in [0, 1]$) quantifies the response kinetic of a visual cell during the first step response of the "chirp" stimulus. Cells with a sustained response characteristics have a RTi $= 0$, whereas transient cells with a response decay back to baseline have RTi $= 1$. The RTi was computed as

$$\text{RTi} = 1 - \frac{x[\text{peak} + a]}{x[\text{peak}]}, \tag{8}$$

where *peak* defined the time point of peak response and $a = 400\,\text{ms}$ the response following the peak.

For the RTi, we tested both direct and inverse $(1 + \text{RTi})^{-1}$ relationships and report the maximum of both.

## A.4  Outlier detection

First, the number of neighbors for each point $\mathbf{z}_i$ was calculated within a fixed radius $d = 4$. Neighbors were defined as points $\mathbf{z}_j$ satisfying $\|\mathbf{z}_i - \mathbf{z}_j\| \leq d$, excluding $\mathbf{z}_i$ itself. Points with more than one neighbor ($N_i > 1$) were excluded, as they were considered part of dense regions. Next, KDE was used to estimate the density of the remaining points. Using a Gaussian kernel with bandwidth $h = 0.5$, the density at each point $\mathbf{z}_i$ was calculated as

$$f(\mathbf{z}_i) = \frac{1}{nh^2} \sum_{j=1}^{n} \exp\left(-\frac{\|\mathbf{z}_i - \mathbf{z}_j\|^2}{2h^2}\right), \tag{9}$$

where $n$ is the number of filtered points. Points with log-density values $\log f(\mathbf{z}_i) < -6.0$ were classified as outliers.

### A.5 Allen Institute Neuropixels spiking dataset post-processing

Out of the stimulus battery available in the Neuropixels dataset, we used responses to two stimuli: light flashes (250 ms duration, $n_{\text{trials}} = 75$) and drifting gratings (200 ms duration, $n_{\text{trials}} = 14$, 8 orientations, 5 frequencies), binning the spike trains at a temporal resolution of 25 ms. We grouped neurons by brain area using anatomical annotations from the Allen Institute. Brain areas were the following: posterior accessory optic nucleus (APN), dorsal lateral geniculate nucleus (LGd), ventral lateral geniculate nucleus (LGv), lateral posterior nucleus (LP), unsepcified visual cortex (VIS), anterolateral visual cortex (VISal), anteromedial visual cortex (VISam), lateral visual cortex (VISl), primary visual cortex (VISp), posteromedial visual cortex (VISpm), rostrolateral visual cortex (VISrl) and excluded non-visual areas.

### A.6 Implementation details

TRACE uses a lightweight multi-layer perceptron (MLP) that can run efficiently on a GPU. To ensure a fair comparison, we matched CEED's architecture [42] and used an MLP consisting only of four layers with sizes [768, 512, 256, 128] and ReLU activations between them. We added a 1024-dimensional projection head. Due to TS2Vec's specialized augmentations and hierarchical contrastive scheme, harmonizing its architecture was more difficult and we stuck to its original, dialated CNN architecture.

For the synthetic datasets, we ran all methods with batch size 512 for 100 epochs unless for TS2Vec→PCA, where we kept the default batch size and of number of epochs. We trained TRACE and CEED→PCA with learning rate 0.3 and TS2Vec with its default learning rate.

For the calcium imaging dataset, we used optimal hyperparameters found with a grid search with batch sizes ranging from 1024 to 3200 and learning rates from 0.1 to 0.2 for both TRACE and CEED. The best hyperparameter setting was chosen based on the final loss. For TRACE the best batch size was 1280, while for CEED it was 1024. A learning rate of 0.1 was optimal for both. For TRACE + CEED we used hyperparameters as in the unmodified TRACE version.

For the Neuropixels dataset we ran a separate grid search and identified optimal hyperparameters for for TRACE and CEED (0.03 learning rate, batch size 512) and for TRACE + CEED (0.08 learning rate, batch size 1024).

Due to the much longer run time of TS2Vec such a grid search was infeasible. For a fair comparison, we massively increased the number of epochs to 1000 for TRACE+TS2Vec and increased the batch size to 768 (larger sizes exceeded memory constraints). To stay close to TS2Vec's original implementation for TS2Vec→PCA, we kept the short default runtime ($< 1$ epoch) and batch size 16. The learning rate for TS2Vec remained at its default value of 0.001. We used the same setting for both neural datasets.

Unless otherwise specified, we trained all embeddings for 1000 epochs for the calcium imaging dataset.

Computations were performed on an NVIDIA A40 GPU 48 GB.

### A.7 CEED augmentations

Here we give details on the general-purpose time-series augmentaions that we used when running CEED and TRACE+CEED. We apply these augmentation to the mean of all trials,

$$x_i = \frac{1}{r} \sum_{l=1}^{r} x_i^l.$$

Amplitude jittering randomly scales the signal between 0.7 and 1.3 using a uniform distribution,

$$x_i' = r' \cdot x_i \text{ and } x_i'' = r'' \cdot x_i \quad \text{with } r', r'' \sim \mathcal{U}(0.7, 1.3), \tag{10}$$

while temporal jittering shifts the signal by up to 3 time bins (up to 370 ms for 8 Hz sampling frequency)

$$x_i'[t] = x_i[t - s'] \text{ and } x_i''[t] = x_i[t - s''] \quad \text{for } s', s'' \sim \mathcal{U}(\{\pm 1, \pm 2, \pm 3\}). \tag{11}$$

To efficiently handle the computationally expensive correlated noise generation, we pre-computed 10,000 noise samples from the covariance matrix of the data using a multivariate normal distribution and added them to the original time series, mimicking the augmentation

$$x_i' = x_i + \varepsilon' \text{ and } x_i'' = x_i + \varepsilon'' \quad \text{for } \varepsilon', \varepsilon'' \sim \mathcal{N}\big(\mathbf{0}, \text{Cov}(\{x_1, \ldots, x_n\})\big). \tag{12}$$

We applied each transformation independently, with probabilities of 0.7 for amplitude jittering, 0.6 for temporal jittering, and 0.5 for correlated noise injection. Hyperparameters of the transformations were adapted to our datasets.

## A.8 Supplementary tables and figures

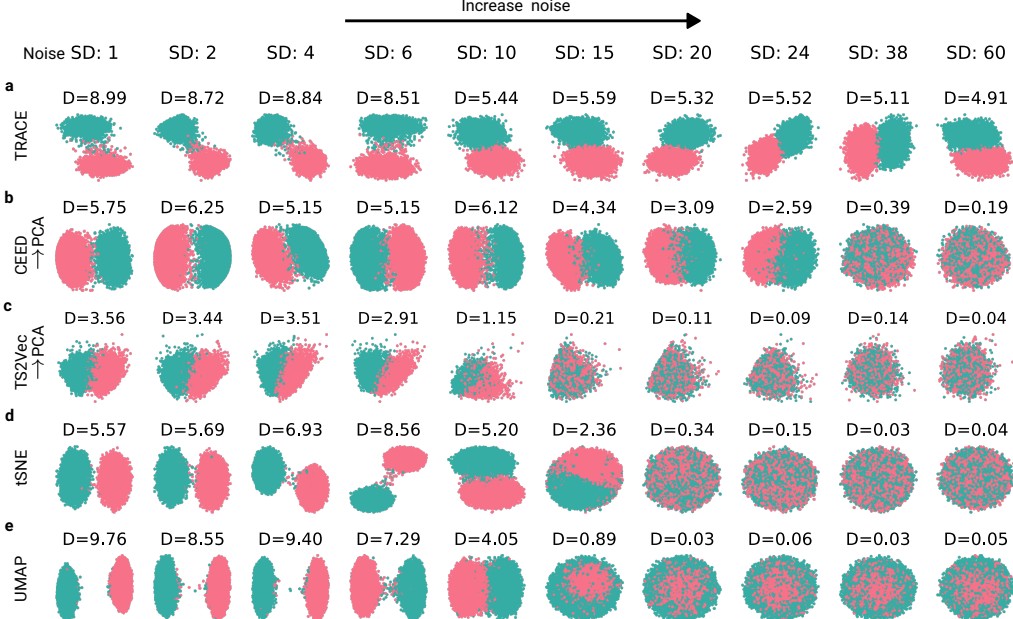

Figure S1: **TRACE is able to separate classes in the synthetic dataset even for high noise levels and outperforms other methods. (a)** TRACE embedding for increasing amounts of noise during the neural baseline activity. Discriminability (D) values indicated at the *top* of each embedding. Standard deviation (SD) of noise is indicated at the *top* of the figure. **(b)** Same as (a), but for CEED → PCA, **(c)** TS2Vec → PCA, **(d)** $t$-SNE, and **(e)** UMAP.

Table S1: Mean discriminability for simulated responses across three seeds for different noise levels (standard deviation of baseline response). Best performing method in **bold**.

| Noise SD | 1 | 2 | 4 | 6 | 10 | 15 | 20 | 24 | 38 | 60 |
|---|---|---|---|---|---|---|---|---|---|---|
| TRACE | 9.01 | **8.75** | 8.87 | 6.93 | **5.57** | **5.57** | **5.45** | **5.47** | **5.14** | **3.52** |
| CEED → PCA | 6.01 | 6.13 | 5.13 | 4.76 | 5.26 | 3.84 | 2.98 | 1.90 | 0.37 | 0.18 |
| TS2Vec → PCA | 3.65 | 3.61 | 3.68 | 3.16 | 1.24 | 0.22 | 0.11 | 0.10 | 0.13 | 0.05 |
| $t$-SNE | 5.56 | 5.65 | 6.94 | **8.40** | 5.24 | 2.35 | 0.45 | 0.09 | 0.03 | 0.03 |
| UMAP | **9.76** | 8.55 | **9.40** | 7.29 | 4.05 | 0.89 | 0.03 | 0.06 | 0.03 | 0.05 |

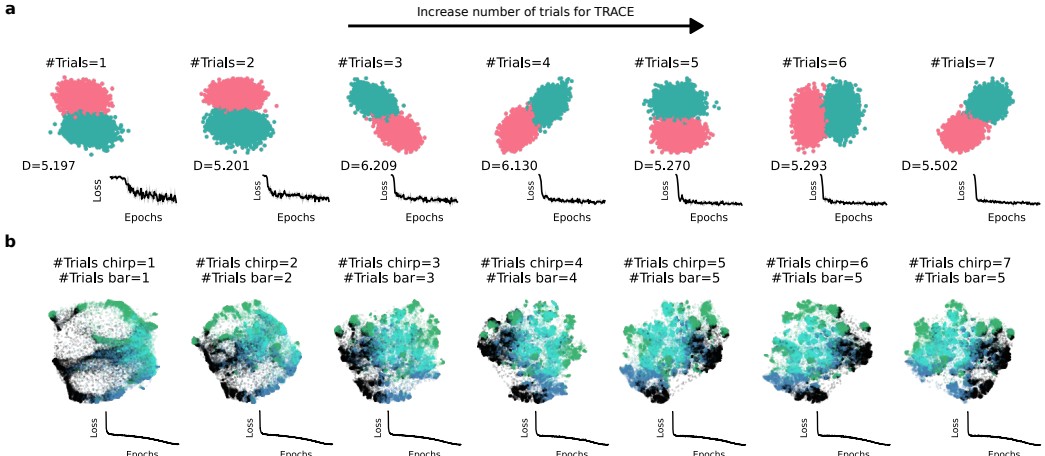

Figure S2: **TRACE embeddings with varying numbers trials** $k$ **for averaging. (a)** *From left to right:* Embeddings for the synthetic dataset with increasing numbers of trials $k$ from 1 to 7 used for the non-overlapping subsets. Discriminability (D) between class 1 and class 2 neurons was computed as in Fig. 2. *Bottom:* Loss over epochs. **(b)** *From left to right:* Embeddings for the calcium imaging dataset with increasing numbers of trials per subset mean $k$ from 1 to 7 for the "chirp" stimulus and from 1 to 5 for the moving bars stimulus. Neurons are colored by the four broad groups OFF (blue), ON-OFF (turquoise), ON (green), Suppressed-by-contrast (black). *Bottom:* Loss over epochs.

Table S2: Reducing number of trials ($k$) used for the non-overlapping subsets using the calcium imaging dataset. The maximum number of trials was 7 for the "chirp" stimulus ($k_{chirp}$) and 5 for the moving bar stimulus ($k_{moving\ bar}$). Best performance in **bold**.

| $k_{chirp}$ | 1 | 2 | 3 | 4 | 5 | 6 | 7 |
|---|---|---|---|---|---|---|---|
| $k_{moving\ bar}$ | 1 | 2 | 3 | 4 | 5 | 5 | 5 |
| ARI | 0.24 | 0.27 | 0.27 | 0.28 | **0.29** | 0.28 | **0.29** |
| $k$NN accuracy | 51.9 | 62.4 | 68.1 | 68.9 | 68.6 | 69.3 | **69.5** |
| $r_S$ | **0.48** | 0.43 | 0.45 | 0.45 | 0.42 | 0.47 | 0.47 |
| $r_{OOi}$ | **0.71** | 0.69 | 0.67 | 0.55 | 0.64 | 0.70 | 0.67 |
| $r_{RTi}$ | **0.30** | 0.24 | 0.28 | 0.27 | 0.29 | **0.30** | 0.24 |
| $r_{Depth}$ | 0.58 | **0.62** | 0.58 | 0.48 | 0.55 | 0.57 | 0.54 |

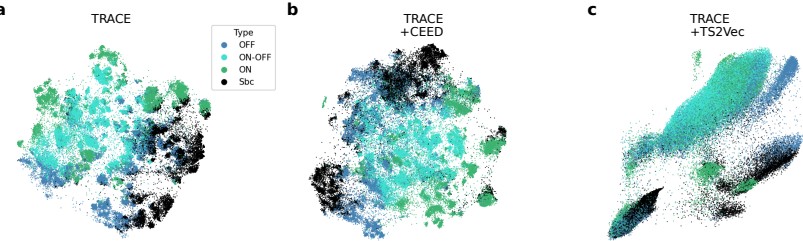

Figure S3: **TRACE-variants visually exhibit a good balance between resolving finer cluster structure and retaining large-scale structure of the calcium imaging dataset. (a – c)** Two-dimensional embeddings of (a) TRACE, (b) TRACE+CEED, (c) TRACE+TS2Vec. Color-coded according to their functional group.

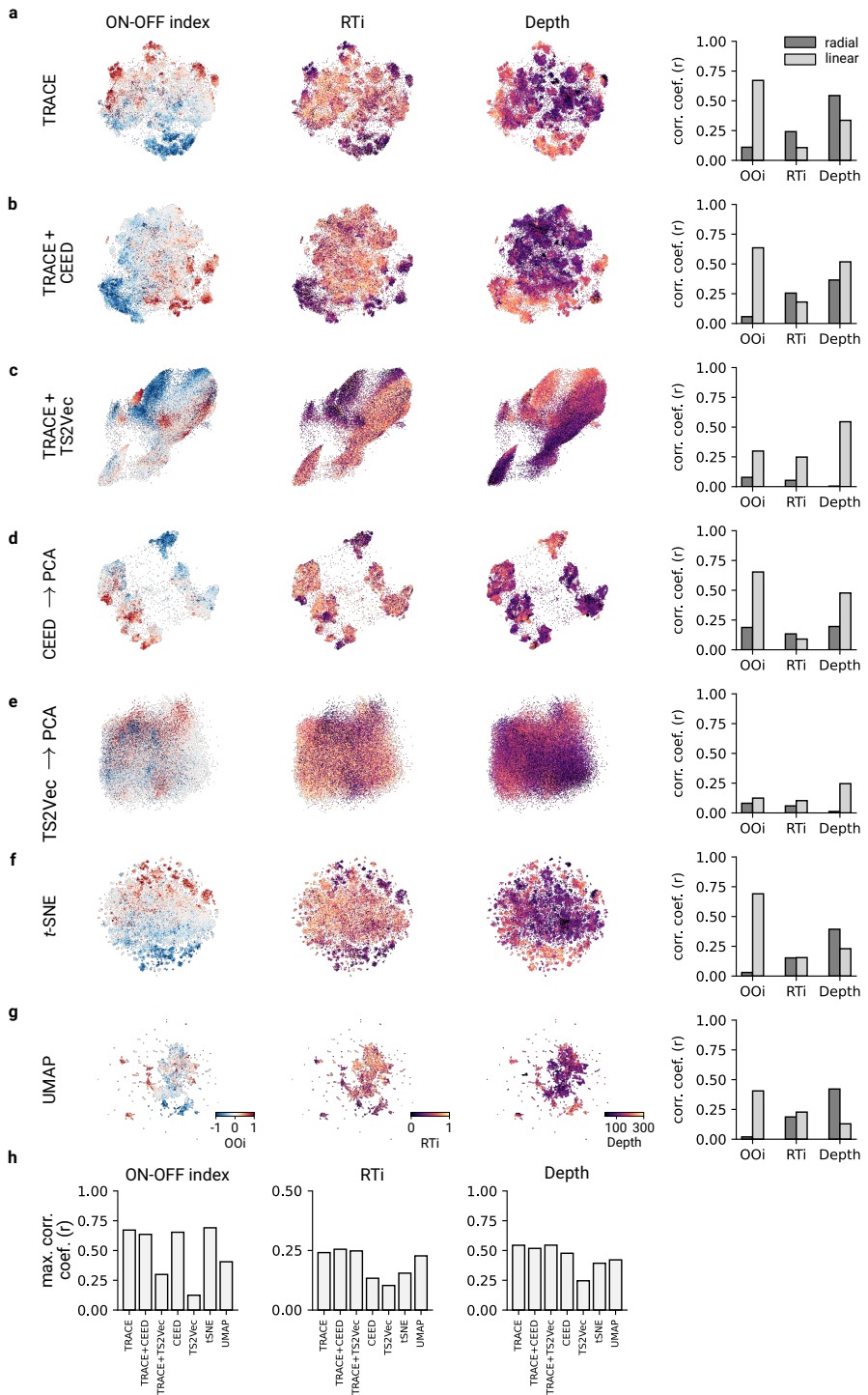

Figure S4: **Evaluating low dimensional embeddings of the calcium imaging dataset by their correlation with biological variables. (a)** TRACE embedding colored by the ON-OFF-index (OOi, *left*), response transience index (RTi, *mid-left*), and recording depth in the superior colliculus (*mid-right*). *Right*: Absolute Pearson correlation coefficients between TRACE embedding and the three biological variables using radial and linear transformations to correlate. **(b)** Same as (a), but using the TRACE + CEED embedding, **(c)** TRACE + TS2Vec, **(d)** CEED→PCA, **(e)** TS2Vec→PCA, **(f)** *t*-SNE, and **(g)** UMAP. **(h)** Maximum correlation coefficients for each embedding for OOi (*left*), RTi (*middle*), and depth (*right*).

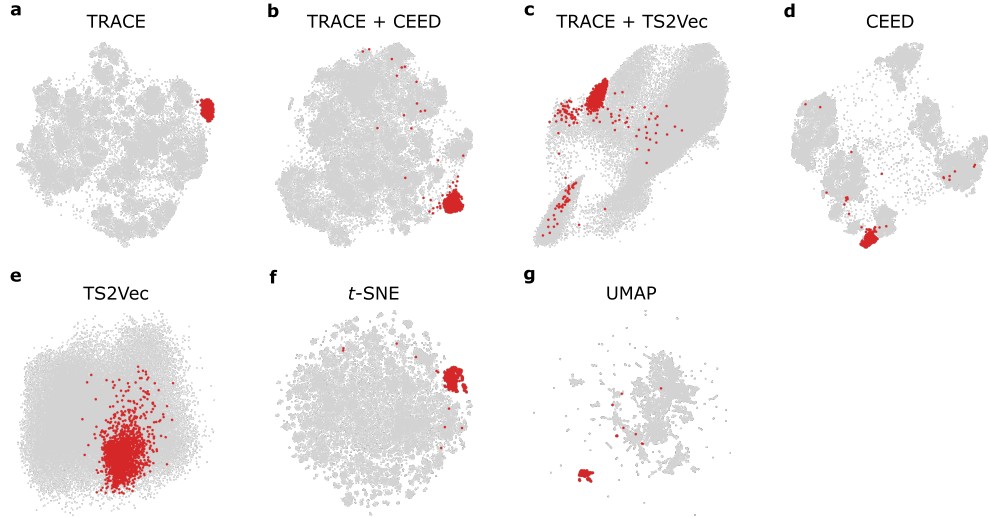

Figure S5: **Identified artifacts in the calcium imaging dataset across all methods. (a)** TRACE embedding with identified artifact island in red. **(b)** Same as in (a) but for TRACE + CEED, **(c)** TRACE + TS2Vec, **(d)** CEED → PCA, **(e)** TS2Vec → PCA, **(f** $t$-SNE, and **(g)** UMAP.

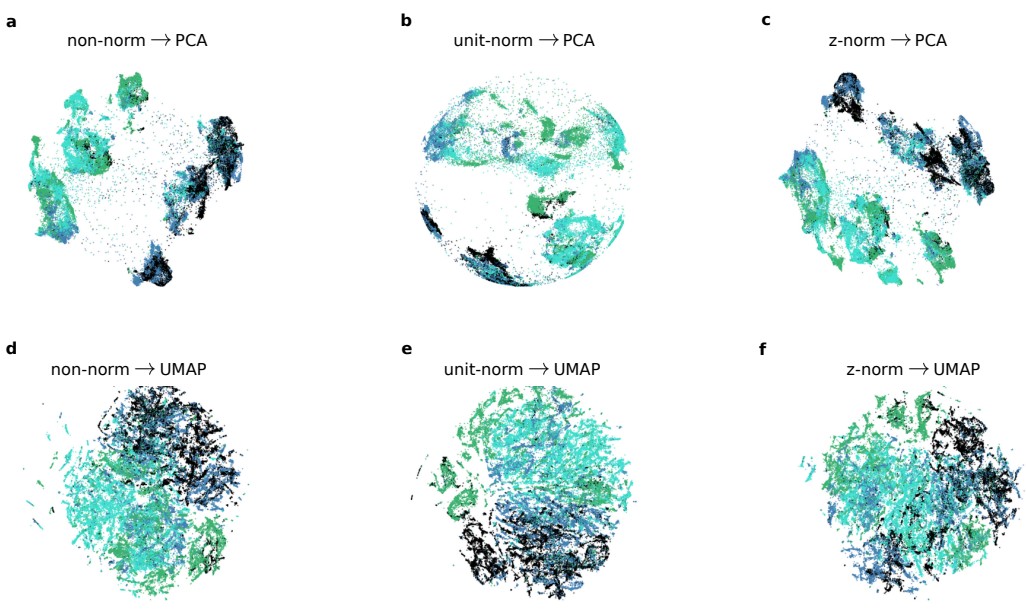

Figure S6: **Different dimensionality reduction options for visualization of CEED's 5D output in 2D space. (a – c)** 2D PCA visualizations of the original CEED outputs in $\mathbb{R}^5$ (a), CEED outputs normalized to the hypersphere $S^4 \subset \mathbb{R}^d$ (b), the 5D CEED output with each dimension scaled to unit variance (c). **(d – f)** 2D UMAP visualizations of the same 5D data as in (a–c).

Table S3: Evaluating dimensionality reduction options for visualizing CEED's 5-dimensional embeddings in 2D space. For the different normalizations, see the caption of Fig. S6. Best performance in **bold**. All metrics other than the aggregated rank are better when higher.

| Model | $r_{OOi}$ | $r_{RTi}$ | $r_{Depth}$ | $\mu_{Rank}$ |
|---|---|---|---|---|
| CEED (non-norm PCA) | 0.58 | 0.14 | 0.54 | **2.33** |
| CEED (non-norm UMAP) | 0.36 | **0.19** | 0.54 | 3 |
| CEED (unit-norm PCA) | 0.55 | 0.11 | **0.55** | 3 |
| CEED (unit-norm UMAP) | 0.38 | 0.16 | 0.31 | 4.33 |
| CEED (z-norm PCA) | **0.59** | 0.12 | 0.53 | 3 |
| CEED (z-norm UMAP) | 0.42 | 0.09 | 0.39 | 5 |

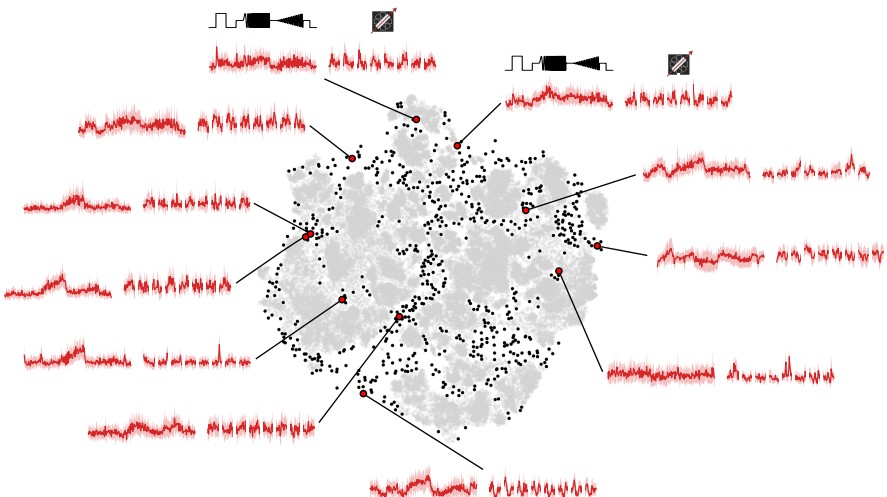

Figure S7: **Outliers in the TRACE embedding of retinal output neurons are noisy.** Isolating and detecting outliers (black) in the TRACE embedding space (gray). Example outliers are highlighted in red showing their highly noisy "chirp" and moving bar responses.

Table S4: Results for different methods on the calcium image data for varying embedding dimension. TRACE performed best in terms of $k$NN accuracy in all dimensions and maintained its performance in terms of Spearman correlation behind CEED, while TS2Vec degraded significantly. TS2Vec used the cosine similarity for $d > 2$.

| Model | $k$NN accuracy | | | | Spearman correlation $r_S$ | | | |
|---|---|---|---|---|---|---|---|---|
| | $d=2$ | $d=16$ | $d=128$ | $\mu_{Rank}$ | $d=2$ | $d=16$ | $d=128$ | $\mu_{Rank}$ |
| TRACE | **69.7** | **75.3** | **75.5** | **1** | 0.45 | 0.4 | 0.41 | 2.33 |
| CEED | 64.0 | 72.3 | 72.2 | 2 | 0.51 | **0.58** | **0.51** | **1.33** |
| TS2Vec | 23.8 | 54.9 | 66.4 | 3 | **0.60** | 0.13 | 0.18 | 2.33 |

Table S5: Training time for all methods on the calcium imaging dataset. Data augmentations for CEED and TRACE + CEED were pre-computed and sampled from during training to improve training time (see Sec. A.7). Please note that no data augmentations are necessary for TRACE. Each experiment was run on a NVIDIA A40 GPU 48 GB. For implementation details, see Sec. A.6.

| Model | Epochs | Time (mm:ss) |
|---|---|---|
| TRACE | 1000 | $23:17 \pm 11.8s$ |
| TRACE + CEED | 1000 | $25:40 \pm 5.1s$ |
| TRACE + TS2Vec | 1000 | $962:43 \pm 2668.8s$ |
| CEED → PCA | 1000 | $25:17 \pm 19.7s$ |
| TS2Vec → PCA | <1 | $1:23 \pm 2.0s$ |
| $t$-SNE | 1000 | $6:59 \pm 9.4s$ |
| UMAP | 1000 | $4:56 \pm 8.3s$ |

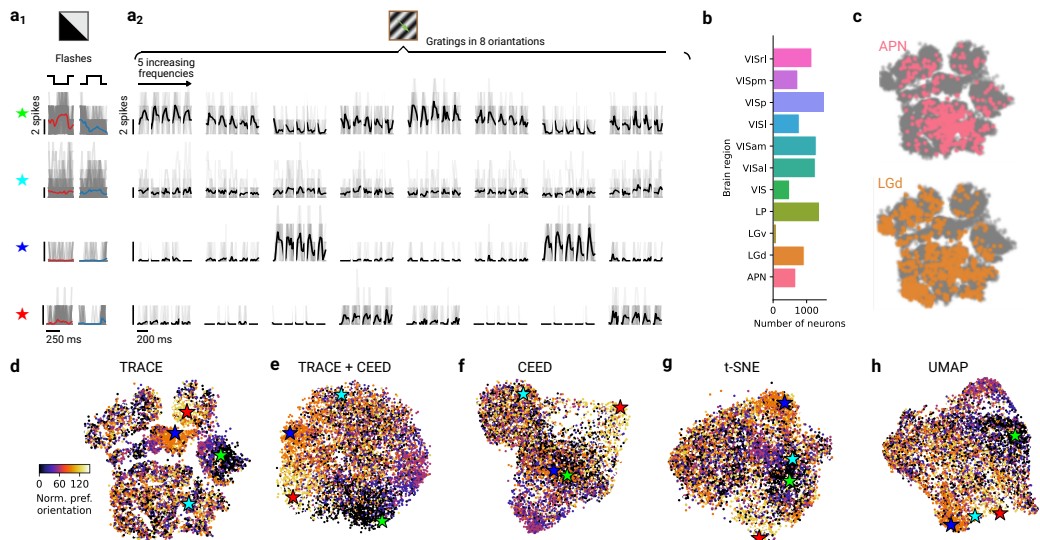

Figure S8: **Neuropixels dataset. (a)** Mean responses to dark flashes (*red*) and bright flashes (*blue*) ($a_1$), as well as drifting gratings ($a_2$, *black*) for four example neurons. Single trial responses in *gray*. **(b)** Number of neurons per brain area. **(c)** Coverage of two example brain areas (APN, LGd) in the TRACE embedding. **(d)** TRACE embedding of the neuronal time series color-coded according to their preferred orientation. Example neurons from (a) are marked with stars. **(e)** Same as (d), but for TRACE+CEED, **(f)** CEED visualized in 2D using PCA, **(g)** $t$-SNE, and **(h)** UMAP.

Table S6: **Quantitative model performance on the Neuropixels dataset**. The columns are: model type (see text), the ARI score, $k$NN accuracy, Spearman correlation ($r_S$), correlations with biologically meaningful variables (orientation selective index (OSI), grating modulation ratio (F1/F0), natural image selectivity (NIS), $k$NN regression for preferred orientation (PO), and mouse behavior, and the average rank $\mu_{\text{Rank}}$ of each method. For a definition of the measures, see Sec. 4.2. All metrics other than the rank are better when higher. Uncertainties were insignificant and we omitted them. The best values for each metric are **bold**.

| Model | ARI | $k$NN acc | $r_S$ | OSI | PO | F1/F0 | NIS | behavior | $\mu_{\text{Rank}}$ |
|---|---|---|---|---|---|---|---|---|---|
| TRACE | **0.04** | **23.7%** | 0.40 | 0.38 | **0.10** | 0.17 | 0.12 | **0.11** | **1.8** |
| + CEED | 0.03 | 20.7% | 0.21 | 0.35 | 0.02 | 0.09 | 0.08 | 0.03 | 3.5 |
| + TS2Vec | 0.01 | 16.9% | 0.03 | 0.36 | 0.01 | **0.29** | 0.25 | 0.01 | 4.5 |
| CEED → PCA | 0.03 | 18.9% | 0.20 | 0.27 | 0.01 | 0.09 | 0.05 | 0.07 | 4.3 |
| TS2Vec → PCA | 0.03 | 18.5% | 0.04 | **0.41** | 0.01 | 0.14 | **0.29** | 0.02 | 4.0 |
| $t$-SNE | 0.01 | 21.8% | 0.50 | 0.23 | 0.09 | 0.06 | 0.03 | 0.01 | 4.5 |
| UMAP | 0.01 | 18.6% | **0.52** | 0.13 | 0.06 | 0.03 | -0.02 | 0.01 | 5.5 |

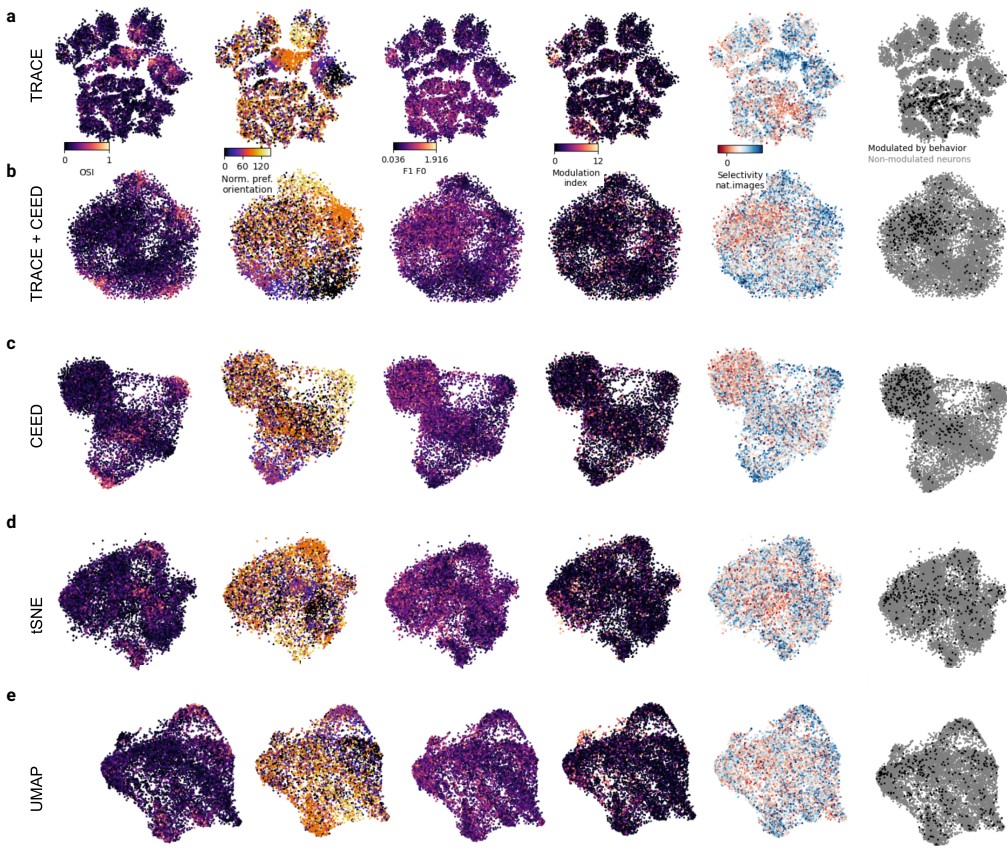

Figure S9: **TRACE finds biologcially relevant structure in Neuropixels data set. (a)** TRACE embedding colored by the OSI, the normalized preferred direction, the F1/F0 ratio, the modulation index, the selectivity index for natural images, and the neurons that are modulated by behavior vs. non-modulated by behavior. **(b)** Same as in (a) but for TRACE + CEED, **(c)** CEED → PCA, **(d)** $t$-SNE, and **(e)** UMAP.

