# OpenReview forum: "TRACE: Contrastive learning for multi-trial time series data in neuroscience"
_NeurIPS.cc/2025/Conference — NeurIPS 2025 poster_

### Official Review · Reviewer_1qve · 2025-06-25

**Clarity:** 3
**Significance:** 3
**Originality:** 3
**Rating:** 4
**Confidence:** 2

**Summary:**

The authors propose a novel method, TRACE, for projecting neurons in a 2D space based on their multi-trial activity using contrastive learning. Using cauchy kernels they force the model to project to a 2D space and not in a hyper-sphere as it is usually done. As every contractive method positive and negative pairs are needed. The positive pairs are the same neuron averaged across different trials, while the negative pairs are different neurons across different trials. The method is evaluated on simulated data and 2photon imaging neural data from mice. In their results section, they show that their TRACE is able to cluster neurons based on their activity patterns better than other methods such as CEED, TS2VEC, tSNE, and UMAP.

**Questions:**

Questions:
- I found subsection 3.2 a bit unclear. Specifically, how many positive pairs are created per neuron during training? Are the subsets fixed or dynamically resampled?

**Ethical Concerns:**

["NO or VERY MINOR ethics concerns only"]

**Final Justification:**

As I wrote from my initial review, I found that this paper is technically solid and is interesting for the field. The author's rebuttal further clarified some points, and I am pleased to keep my initial score.

**Limitations:**

yes

**Paper Formatting Concerns:**

No problems with the paper formatting

**Quality:**

4

**Strengths And Weaknesses:**

Strengths:
- The method addresses a real problem in neuroscience, which is the need to visualize and interpret high-dimensional neural data.
- If training the MLP model and choosing the right hyper-parameters is straightforward, the method could be widely applicable to many datasets.
- Using this method as an outlier / artifact detection could be potential useful.

Weaknesses:
- A potential limitation is that neurons with highly variable or even contradictory responses across trials (e.g., responding positively in some trials and negatively in others) could result in positive pairs that are poorly correlated or even misleading. This raises the question of whether the subset-averaging strategy could inadvertently wash out meaningful variability or create noisy positive samples.

---

> ### Author Rebuttal · Authors · 2025-07-30
>
> We thank the reviewer for their positive assessment of our work and the interesting questions.
>
> **(1) Subset-averaging strategy:** We acknowledge the reviewers concern about neurons with highly variable or contradictory responses across trials potentially creating misleading positive pairs. TRACE operates under the assumptions that differences between trials are (a) undesirable noise but (b) characteristic for the local noise structure between different neurons. In some cases, between-trial variation may be of interest, e.g., when investigating adaptation over multiple trials. This violates our assumption (a) and TRACE is not the right tool for this type of question. Conversely, if between-trial variation is not of interest and some neuron responds very differently in different trials, this may simply mean that responses are very noisy for this and similar neurons. The resulting diverse positive samples are not misleading, but rather correctly teach TRACE to strongly denoise this part of the data.
> In fact, TRACE's data trial-driven positive pairs may be less likely to "wash out" meaningful difference than other methods, which operate on averages across all trials and thus do not incorporate any information about the between-trial variation.
> Standard approaches like t-SNE and UMAP compute Euclidean distances after averaging across all trials, which can incorrectly bias them by uninformative parts of the time series (Sec 5.1). CEED applies generic data augmentations that may or may not be aligned to the true local noise structure of the data.
> We think this point raised by the reviewer is important and we will incorporate this discussion into Sec 6 of our manuscript for the revision.
>
> **(2) Generation of positive samples:** We thank the reviewer for pointing out the insufficient description of how we generate positive samples. To clarify, during training, each mini-batch contains subset means from multiple neurons. For each neuron, we create one positive pair consisting of two time series. To create one part of the pair, a random subset of trials of that neuron is selected and averaged. To create the second part of the pair, another non-overlapping subset is averaged. One such pair of subsets is dynamically resampled for each neuron per epoch. We will describe this positive sample generation in a revised Sec 3.2 and the code will also be made publicly available.
>
>
> We hope this resolves your concerns. Please let us know if you have any concerns left.

---

> > ### Comment · Reviewer_1qve · 2025-08-04
> >
> > Thank you for your replies and for clarifying/answering my comments/questions, the positive sample generation is much clearer now.
> > I also found interesting that your method does a good job in clustering cortical data (answer to reviewer nFhS).

---

### Official Review · Reviewer_nFhS · 2025-07-02

**Clarity:** 4
**Significance:** 3
**Originality:** 2
**Rating:** 4
**Confidence:** 3

**Summary:**

The paper introduces TRACE, a contrastive learning framework for visualizing dependencies among multi-variate (e.g., neurons) time-series data over multiple-trials. TRACE generates positive pairs by averaging disjoint subsets of trials and directly learns a 2D embedding using a t-SNE-inspired Cauchy kernel.

The authors compare TRACE to baselines on both simulated and large-scale in vivo neural datasets. For a two-photon imaging dataset of retinal ganglion cell axon endings, TRACE's embeddings capture biologically relevant continuous variations, cell-type-related cluster structures, and can assist in identifying recording artifacts.

**Questions:**

None

**Ethical Concerns:**

["NO or VERY MINOR ethics concerns only"]

**Final Justification:**

The authors' rebuttal resolved my concerns regarding limited experimental analysis with a single dataset and potential impact on few application domains.

**Limitations:**

yes

**Quality:**

3

**Strengths And Weaknesses:**

The problem setting, method and experiments of the submitted manuscript are well written and very clear.
The authors provide code and detailed results for the considered baselines in the appendix.

Weaknesses:

 - Quality: the authors limit their analysis to a single neuroimaging dataset despite a plethora of public multi-trail, multi-variate neuroimaging datasets.
 - Originality: the authors develop TRACE around the time-tested idea of averaging responses across multiple repetitions to improve the SNR, which is a standard technique in neuroscience. They then use boosting to sample positive pairs and standard ML techniques like a contrastive loss with a Cauchy kernel.
 - Significance: TRACE requires multiple repetitions and the assumption of independent, additive noise (to improve the SNR via averaging repetitions). This limits TRACE to specific applications like neuro imaging with likely no impact on other ML application domains.

---

> ### Author Rebuttal · Authors · 2025-07-30
>
> **(1) Additional neural data set**: Based on the reviewer's comment, we applied TRACE to an additional neural dataset with spike responses: the Allen Institute neuropixels 'visual coding' dataset (de Vries et al. 2020; Siegel et al. 2021). It's the largest publicly available dataset of spiking responses across visual brain areas, containing 10,322 neurons (after quality filtering, excluding non-visual neurons) from awake and behaving mice responding to light flashes (450 ms, n_trials=75) and drifting gratings (n_trials=14, 250 ms, 8 orientations, 5 frequencies). We applied TRACE and other methods to binned spike trains at a resolution of 25 ms. The spike train data consists of discrete neural events that follow Poisson-like statistics, where the timing and occurrence of spikes carry important information. TRACE is especially beneficial for such spiking data because preserves the natural statistical structure through trial subset averaging, whereas standard augmentations used by contrastive methods (Gaussian noise, temporal jitter) add continuous noise that destroys the discrete nature of spike trains.
> We evealuated the performance using standard clustering metrics (ARI, kNN accuracy, Spearman distance correlation) with brain region as class labels and biologically meaningful variables including orientation selective index (OSI), preferred orientation (PO), grating modulation ratio (F1/F0), natural image selectivity (NIS; based responses to natural images not used to create the 2D embedding), behavioral modulation (correlation of firing rate with running speed of the mouse; not used to create the 2D embedding).
> Alignement of the embedding with biological variables was measures with kNN regression ($R^2$). For PO and behavior, we computed the radial correlation, as it better captured their global structure in the embedding. We also compute the average rank each method had per metric.
> While standard methods (t-SNE, UMAP) found some structure especially in terms of distance correlation and for OSI, TRACE outperformed them in most metrics and CEED in all of them. The performance of TS2Vec variants was mixed, with good preservation of some structure (OSI, NIS) but very poor global distance correlation. Indeed, the global structure of the TRACE+TS2Vec was highly inconsistent across runs. Overall, TRACE achieved the best rank by far, indicating best overall performance on this dataset. Particularly noteworthy is TRACE's ability to identify image selectivity and modulation by behavior without this information being included during training.
> Here we report the mean across three seeds:
> | Model | ARI | *k*NN acc | $r_S$ | OSI | PO | F1F0 | NIS | behavior | Rank |
> |-------|-----|-----------|-----------------|-----|----|----- |-----|----------|------|
> | TRACE | **0.04** | **23.7%** | 0.40 | 0.38 | **0.10** | 0.17 | 0.12 | **0.11** | **1.8** |
> | &nbsp;&nbsp;&nbsp;&nbsp;+ CEED | 0.03 | 20.7% | 0.21 | 0.35 | 0.02 | 0.09 | 0.08 | 0.03 | 3.5 |
> | &nbsp;&nbsp;&nbsp;&nbsp;+ TS2Vec | 0.01 | 16.9% | 0.03 | 0.36 | 0.01 | **0.29** | 0.25 | 0.01 | 4.5 |
> | CEED | 0.03 | 18.9% | 0.20 | 0.27 | 0.01 | 0.09 | 0.05 | 0.07 | 4.3 |
> | TS2Vec | 0.03 | 18.5% | 0.04 | **0.41** | 0.01 | 0.14 | **0.29** | 0.02 | 4.0 |
> | *t*-SNE | 0.01 | 21.8% | 0.50 | 0.23 | 0.09 | 0.06 | 0.03 | 0.01 | 4.5 |
> | UMAP | 0.01 | 18.6% | **0.52** | 0.13 | 0.06 | 0.03 | -0.02 | 0.01 | 5.5 |
>
> We believe this additional validation on a large-scale dataset addresses the reviewer's concern about generalization and demonstrates TRACE's broad applicability across neuroscience dataset types.
>
> **(2) Novelty of the approach**: While we acknowledge that some individual components of our method existed before, we respectfully argue that our core contribution lies in the novel idea of creating positive pairs by subset means over trials. Moreover, TRACE is the first method to combine Cauchy kernel and self-supervision to address specific challenges in neuroscience data visualization. This is not a common combination, as most contrastive methods use cosine similarity, not suitable for visualization.
> This leads to the following advantages of TRACE over previous methods:
> (1) Noise robustness: Results on simulated data show that standard methods like t-SNE and UMAP are sensitive to noisy baseline activity, while current contrastive approaches like CEED use augmentations that mask subtle biological differences. Only TRACE was able to successfully separate the cell types by combining trial subset averaging with contrastive learning for direct 2D visualizations (Sec. 5.1, Fig 2).
> (2) Artifact detection: We showed that TRACE directly identifies experimental artifacts that would would otherwise remain undetected. The artifact island was only clearly visible in the TRACE embedding (Sec 5.3, Fig. 4).
> Together, these novel contributions make TRACE a valuable tool for the neuroscience community. Current methods either lack the domain-specific design needed for multi-trial data or fail to provide biological interpretability. This gap is addressed through the TRACE framework.
>
> **(3) Role of subset means**: We take subset means to create diverse positive pairs that are still similar to the inference-time inputs, not primarily to increase SNR (see end of Sec 3.2). Indeed, it is important that the positive pairs are still noisy and do *not* have the highest possible SNR, because it is precisely their noise structure to which TRACE learns to become invariant. That is the reason why TRACE also works decently when positive pairs are individual trials (Sec 5.2, Fig S2, Table S2). If we instead computed the average across all trials to maximize the SNR, both members of the positive pair would be identical and not provide any learning signal.
>
> **(4) Generalizability across domains**:
> While we acknowledge that we present TRACE as a framework particularly suitable for neuroscience data, many time-series datasets have an inherent multi-trial structure or can be reshaped into this format. Often, time-series data contains natural repetitions or can be segmented into comparison units.
> To give a couple of examples: In sport analytics inertial measurements are often taken during repeated exercises, such as basketball free-throw drills (Hoelzemann et al. 2023, García-de-Villa et al. 2022). The Google Speech Command Dataset (Warden et al. 2018) contains audio recordings of speech commands with 5 trials per speech command and speaker.
> For medical data such as ectrogardiography one could use different daily cycles or stress tests as trials. For financial market data, trading periods can be used as trials to detect unusual patterns or market anomalies. Another application domain could be climate data where years, seasons, or tidal cycles could be used as trials. We are happy to add this discussion our Sec 6 and hope that this convinces the reviewer that time series data sets from many domains could benefits from TRACE’s ability to identify outliers and subtle pattern differences that standard methods miss.
>
> References: [1] Hoelzemann, A., Romero, J. L., Bock, M., Laerhoven, K. V., and Lv, Q. (2023). Hang-time HAR: A benchmark dataset for basketball activity recognition using wrist-worn inertial sensors. Sensors, 23(13), 5879.
> [2] García-de-Villa, S., Jiménez-Martín, A., and García-Domínguez, J. J. (2022). A database of physical therapy exercises with variability of execution collected by wearable sensors. Scientific Data, 9(1), 266.
> [3] Warden, P. (2018). Speech commands: A dataset for limited-vocabulary speech recognition. arXiv preprint arXiv:1804.03209.
>
> **Hopefully, we could address your concerns and would appreciate you increasing your score!**

---

### Official Review · Reviewer_t9Nx · 2025-07-03

**Clarity:** 2
**Significance:** 2
**Originality:** 2
**Rating:** 4
**Confidence:** 4

**Summary:**

This paper introduces a new model to project neural activity into low(2)-dimensional space for visualization. The proposed method extends on existing latent space models by exploiting contrastive learning across trial repeats in time series data to capture neural functional similarity. The authors tested their method in a retinal ganglion cell dataset and simulated data. The results and comparisons to alternative existing method showed that the method can recover known cell types.

**Questions:**

How does the model perform when using different loss functions? Can the model generalize to time-series data without trial structure? Would the method extend to spiking neural data? How?

**Ethical Concerns:**

["NO or VERY MINOR ethics concerns only"]

**Final Justification:**

I appreciate the authors for the detailed response and experiments. I updated my score accordingly. Adding the experiment on spiking data addressed the question on generalization to other types of datasets. While I acknowledged that the initial motivation for the work was visualization, I suggested including quantitative analysis to i) help emphasize the impact of the tool compared to other methods, and ii) because the presented results had no additional neuroscience insights.

Please, ensure that the new results are included in the final version of the manuscript.

**Limitations:**

The authors list the method limitations in terms of increase computational demands and requirement for trial repeats. Still, a larger limitation is that the method assumes known and single stimulus presentation, when in practice neuroscience experiments present multiple stimuli or external and internal conditions at once. Additionally, the dimensionality of the latent space seems to be limited to a 2D space, regardless of the inherent variability in the data.

**Paper Formatting Concerns:**

No concerns.

**Quality:**

2

**Strengths And Weaknesses:**

The work is clearly presented and provides qualitative evidence that the proposed method outperforms commonly used non-linear latent space methods on a calcium imaging dataset. The authors further experiment to understand the model limitations in trial repeats.

While the main application of the method is visualization, presenting a more quantitative description of the model performance is needed. Moreover, the method is tested in a single neural dataset where neural types are known and have been previously identified. It would be relevant to extend the experiments to other and more complex datasets to evaluate the relevance and generalization of the proposed method, given that alternative methods can also discriminate the studied cell types.

---

> ### Author Rebuttal · Authors · 2025-07-30
>
> **(1) Additional neural data set**: Based on the reviewer's comment, we applied TRACE to an additional neural dataset with spike responses: the Allen Institute neuropixels 'visual coding' dataset (de Vries et al. 2020; Siegel et al. 2021). It's the largest publicly available dataset of spiking responses across visual brain areas, containing 10,322 neurons (after quality filtering, excluding non-visual neurons) from awake and behaving mice responding to light flashes (450 ms, n_trials=75) and drifting gratings (n_trials=14, 250 ms, 8 orientations, 5 frequencies). We applied TRACE and other methods to binned spike trains at a resolution of 25 ms. The spike train data consists of discrete neural events that follow Poisson-like statistics, where the timing and occurrence of spikes carry important information. TRACE is especially beneficial for such spiking data because preserves the natural statistical structure through trial subset averaging, whereas standard augmentations used by contrastive methods (Gaussian noise, temporal jitter) add continuous noise that destroys the discrete nature of spike trains.
> We evealuated the performance using standard clustering metrics (ARI, kNN accuracy, Spearman distance correlation) with brain region as class labels and biologically meaningful variables including orientation selective index (OSI), preferred orientation (PO), grating modulation ratio (F1/F0), natural image selectivity (NIS; based responses to natural images not used to create the 2D embedding), behavioral modulation (correlation of firing rate with running speed of the mouse; not used to create the 2D embedding).
> Alignement of the embedding with biological variables was measures with kNN regression ($R^2$). For PO and behavior, we computed the radial correlation, as it better captured their global structure in the embedding. We also compute the average rank each method had per metric.
> While standard methods (t-SNE, UMAP) found some structure especially in terms of distance correlation and for OSI, TRACE outperformed them in most metrics and CEED in all of them. The performance of TS2Vec variants was mixed, with good preservation of some structure (OSI, NIS) but very poor global distance correlation. Indeed, the global structure of the TRACE+TS2Vec was highly inconsistent across runs. Overall, TRACE achieved the best rank by far, indicating best overall performance on this dataset. Particularly noteworthy is TRACE's ability to identify image selectivity and modulation by behavior without this information being included during training.
> Here we report the mean across three seeds:
> | Model | ARI | *k*NN acc | $r_S$ | OSI | PO | F1F0 | NIS | behavior | Rank |
> |-------|-----|-----------|-----------------|-----|----|----- |-----|----------|------|
> | TRACE | **0.04** | **23.7%** | 0.40 | 0.38 | **0.10** | 0.17 | 0.12 | **0.11** | **1.8** |
> | &nbsp;&nbsp;&nbsp;&nbsp;+ CEED | 0.03 | 20.7% | 0.21 | 0.35 | 0.02 | 0.09 | 0.08 | 0.03 | 3.5 |
> | &nbsp;&nbsp;&nbsp;&nbsp;+ TS2Vec | 0.01 | 16.9% | 0.03 | 0.36 | 0.01 | **0.29** | 0.25 | 0.01 | 4.5 |
> | CEED | 0.03 | 18.9% | 0.20 | 0.27 | 0.01 | 0.09 | 0.05 | 0.07 | 4.3 |
> | TS2Vec | 0.03 | 18.5% | 0.04 | **0.41** | 0.01 | 0.14 | **0.29** | 0.02 | 4.0 |
> | *t*-SNE | 0.01 | 21.8% | 0.50 | 0.23 | 0.09 | 0.06 | 0.03 | 0.01 | 4.5 |
> | UMAP | 0.01 | 18.6% | **0.52** | 0.13 | 0.06 | 0.03 | -0.02 | 0.01 | 5.5 |
>
> We believe this additional validation on a large-scale dataset addresses the reviewer's concern about generalization and demonstrates TRACE's broad applicability across neuroscience dataset types.
>
> **(2) Dimensionality of the latent space**: TRACE is not inherently limited to 2D and the architecture supports any dimensionality. We focus on 2D because our paper is about visualization which requires 2D output independent of the intrinsic dimensionality of the data.
> Based on the reviewer's comment, we generated representations for 16 and 128 dimensions across three seeds:
>
> *k*NN accuracy:
> | Model | D=2 | D=16 | D=128 | Rank |
> | - | - | - | - | - |
> | TRACE | 69.7 | 75.3 | 75.5 | 1 |
> | CEED | 64.0 | 72.3 | 72.2 | 2 |
> | TS2Vec | 23.8 | 54.9 | 66.4 | 3 |
>
> Spearman correlation:
> | Model | D=2 | D=16 | D=128 | Rank |
> | - | - | - | - | - |
> | TRACE | 0.45 | 0.4 | 0.41 | 2 |
> | CEED | 0.51 | 0.58 | 0.51 | 1 |
> | TS2Vec | 0.60 | 0.13 | 0.18 | 3 |
>
> TRACE outperforms competitor models in higher dimensions in kNN accuracy and places rank 2 for the Spearman correlation. Notably, TRACE maintains consistent performance across all dimensions, while TS2Vec shows significant degradation in higher dimensions for Spearman correlation, demonstrating TRACE's robustness across different dimensionalities.
>
> **(3) Quantitative performance metrics**:
> We appreciate the reviewer's interest in quantitative evaluation. However, we respectfully disagree that our work lacks quantitative performance metrics. We dedicate Section 4.2 specifically to quantitative measures, employing three standard metrics widely used in dimensionality reduction literature: Adjusted Rand Index (ARI), kNN accuracy, and Spearman's rank correlation. For the toy dataset, we additionally provide discriminability in the low-dimensional space (Sec. 5.1, Fig 2).
> Further, there is an important limitation for our biological dataset: Standard machine learning quantitative metrics assume the existence of ground truth labels, which are rarely available in experimental neuroscience. The class labels were derived from clustering the neural responses themselves (using other methods), making standard clustering evaluation metrics potentially circular.
> To address this challenge, we additionally used independent biologically meaningful variables such as the ON-OFF index, response-transience index, and recording depth. Recording depth, in particular, represents an experimental variable not derived from neural responses and thus serves as external validation. The strong correlations we observe between our embeddings and these biologically meaningful measures provide an appropriate assessment of method quality for neuroscience applications.
> We believe our current quantitative evaluation - combining standard dimensionality reduction metrics with biological validation - provides a comprehensive assessment suitable for this applications.
>
> **(4) Generalizability across domains**:
> While we acknowledge that we present TRACE as a framework particularly suitable for neuroscience data, many time-series datasets have an inherent multi-trial structure or can be reshaped into this format. Often, time-series data contains natural repetitions or can be segmented into comparison units.
> To give a couple of examples: In sport analytics inertial measurements are often taken during repeated exercises, such as basketball free-throw drills (Hoelzemann et al. 2023, García-de-Villa et al. 2022). The Google Speech Command Dataset (Warden et al. 2018) contains audio recordings of speech commands with 5 trials per speech command and speaker.
> For medical data such as ectrogardiography one could use different daily cycles or stress tests as trials. For financial market data, trading periods can be used as trials to detect unusual patterns or market anomalies. Another application domain could be climate data where years, seasons, or tidal cycles could be used as trials. We are happy to add this discussion our Sec. 6 and hope that this convinces the reviewer that time-series data sets from many domains could benefits from TRACE’s ability to identify outliers and subtle pattern differences that standard methods miss.
> Additionally, you asked whether TRACE works with other loss functions. Our setup requires a contrastive loss function and we chose the most prominent one, InfoNCE. We plan to explore other options, such as Noise Contrastive Estimation, Negative Sampling, or Triplet loss in future work.
>
> References:
> **[1]** de Vries, Saskia EJ, et al. "A large-scale standardized physiological survey reveals functional organization of the mouse visual cortex." Nature neuroscience 23.1 (2020): 138-151.
> **[2]** Siegle, Joshua H., et al. "Survey of spiking in the mouse visual system reveals functional hierarchy." Nature 592.7852 (2021): 86-92. **[3]** Hoelzemann, Alexander, et al. "Hang-time HAR: A benchmark dataset for basketball activity recognition using wrist-worn inertial sensors." Sensors 23.13 (2023): 5879. **[4]** García-de-Villa, Sara, Ana Jiménez-Martín, and Juan Jesús García-Domínguez. "A database of physical therapy exercises with variability of execution collected by wearable sensors." Scientific Data 9.1 (2022): 266. **[5]** Warden, Pete. "Speech commands: A dataset for limited-vocabulary speech recognition." arXiv preprint arXiv:1804.03209 (2018).
>
> **Hopefully, we could address your concerns and would appreciate you increasing your score!**

---

> > ### Comment · Reviewer_t9Nx · 2025-08-01
> >
> > I appreciate the authors for the detailed response and experiments. I updated my score accordingly. Adding the experiment on spiking data addressed the question on generalization to other types of datasets. While I acknowledged that the initial motivation for the work was visualization, I suggested including quantitative analysis to i) help emphasize the impact of the tool compared to other methods, and ii) because the presented results had no additional neuroscience insights.
> >
> > Please, ensure that the new results are included in the final version of the manuscript.

---

> > > ### Author Response · Authors · 2025-08-01
> > >
> > > We thank the reviewer for their reply and appreciating our efforts. We will certainly include the new results in the final manuscript.

---

### Note · Authors · 2025-08-12

**Final remarks:**\
We are pleased that our work received positive evaluation from the reviewers, who noted that *the work is clearly presented and provides qualitative evidence that the proposed method outperforms commonly used non-linear latent space methods* and that our *method addresses a real problem in neuroscience, which is the need to visualize and interpret high-dimensional neural data*.

**Summary of the rebuttal:**\
We added additional experiments, addressing all major concerns of the reviewers.\
**(1)** We validated TRACE on the Allen Institute neuropixels spiking dataset. As hypothesized, TRACE proved particularly beneficial for discrete neural events following Poisson-like statistics, achieving the best overall performance rank across all metrics. This experiment directly addresses generalizability concerns of reviewers t9Nx and nFhS.\
**(2)** We also ran additional experiments to clarify that TRACE is not limited to 2D visualizations and that TRACE performs well in higher dimensions (16D and 128D).

**Changes made:**\
Based on reviewer feedback, we will incorporate: **(1)** the spiking dataset and results, **(2)** performance in higher-dimensions, **(3)** discussion of cross-domain generalizability, and **(4)** clearer description of our positive sample generation.
 \
\
\
We thank the reviewers for their thoughtful feedback and are happy that we were able to address all raised concerns demonstrating that TRACE closes a critical gap for visualizing high-dimensional neuroscience data by naturally exploiting their multi-trial and statistical structure.

---

### Decision · Program_Chairs · 2025-09-17

**Decision:**

Accept (poster)

**Comment:**

There was a consensus that the proposed contrastive learning framework for visualiziing and clustering neural signals, which exploits the repeated trial structure to generate positive pairs, addresses a practical need. The additional Allen spiking neuron experiment during the rebuttal directly addressed reviewer concerns and demonstrated the method's effectiveness on a compelling application. This dataset contains more than 10,000 neurons in response to discrete visual stimuli categories. Thanks to the typical design of a traditional neural coding experiment, their proposed method has a definite edge and outperforms in all comparisons. The method is novel, straightforward, and effective in exploiting the repeasted structure in many neuroscience experiments, but it also limits its applicability to other domains (with some exceptions in exercise and speech) as is. I receommend acceptance because of its impressive performance, usefulness, and potential impact to the neuroscience community.